# Understanding BYOL and SimSiam Under The Lens Of Mutual Information Maximization

## Abstract

We study teacher-student (TS) self-supervised learning methods equipped with a prediction head (e.g., BYOL, SimSiam), which learn meaningful representations without relying on negative samples. Building on the InfoMax perspective that unifies many multi-view Self-Supervised Learning (SSL) families, we show that TS-SSL implicitly maximizes a lower bound on the mutual information $I(Z_\theta; X)$ between the inputs $X$ and the teacher representations $Z_\theta$. Concretely, we prove that, assuming an optimal predictor, BYOL and SimSiam's loss is an approximation $H(Z_\theta \mid Z_\phi, X)$. Building on this results, we prove that, under a mild assumption, verified empirically on six different datasets, the alternating optimization—student prediction (with stop-gradient) followed by teacher updates—implicitly optimizes $\theta$ so that it maximizes $I(Z_\theta; (X, Z_\phi))$ a lower bound on $I(Z_\theta; X)$. Then, we derive increment convergence dynamics of the teacher representation's entropy and alignment during training. Eventually, motivated by these theoretical insights, we introduce a simple mutual-information–based regularizer on the student latent space that enforces monotonic growth of $I(Z_\theta; X)$ and yields consistent downstream improvements on both natural-image and medical-imaging benchmarks.

## 1 Introduction

Representation learning seeks to build compact representations of high-dimensional data that preserve their underlying semantic structure. A central principle guiding this objective is the maximization of mutual information (MI) between inputs and learned representations, ensuring that the latter retain informative content from the data Linsker (1988); Belghazi et al. (2018); Hjelm et al. (2019). MI has provided a unifying perspective across diverse methods, including information bottleneck approaches Tishby et al. (2000); Alemi et al. (2022), contrastive learning Oord et al. (2019); Wang & Isola (2020), deep clustering Caron et al. (2020); Rodríguez-Gálvez et al. (2023), variance–invariance frameworks Zbontar et al. (2021); Bardes et al. (2021); Shwartz-Ziv et al. (2023), masked image modeling Huang et al. (2025), and multi-modal representation learning Dufumier et al. (2025). Crucially, the way MI is maximized may induce inductive biases in learned representations, such as encouraging compressed representations Tishby et al. (2000); Alemi et al. (2022), assuming clustered representations Caron et al. (2018; 2020); Rodríguez-Gálvez et al. (2023), view-invariant representations Wang & Isola (2020), or low-covariance representations Shwartz-Ziv et al. (2023).

A particularly successful paradigm is multi-view self-supervised learning (MVSSL), where the objective enforces agreement between augmented views of the same input Bachman et al. (2019); Tian et al. (2020); He et al. (2020). MVSSL encompasses contrastive approaches Oord et al. (2019); Chen et al. (2020a); He et al. (2020), clustering-based methods Caron et al. (2018; 2020), and, more recently, teacher-student predictor-based frameworks Grill et al. (2020); Chen & He (2021). The connection between MVSSL and mutual information maximization is most explicit in contrastive methods, where the InfoNCE loss can be seen as a lower bound on the MI between representations of different views Oord et al. (2019); Poole et al. (2019). Clustering- and variance–invariance methods can also be interpreted through the MI lens, since they implicitly maximize distinct bounds with specific inductive biases on the learned representations Caron et al. (2018; 2020); Rodríguez-Gálvez et al. (2023); Zbontar et al. (2021); Bardes et al. (2021).

Among MVSSL methods, teacher–student frameworks, such as BYOL Grill et al. (2020) and Sim-Siam Chen & He (2021), differ from other SSL methods in achieving competitive performance without negatives, prototypes, or explicit variance regularization. They rely on an asymmetric architecture combining a predictor, a stop-gradient, and in BYOL, an exponential moving average (EMA) of the teacher network. Despite their empirical success, the mechanisms underlying their stability remain elusive: while contrastive and clustering methods have been tied to MI maximization, BYOL and SimSiam appear to avoid collapse without explicitly maximizing entropy Richemond et al. (2020); Tian et al. (2021); Halvagal et al. (2023). This raises a central open question:

*Can teacher–student distilled SSL methods equipped with a prediction head be understood from the lens of a mutual-information ?*

In this paper, we show that teacher–student SSL, like BYOL and SimSiam, implicitly maximizes a lower bound on the mutual information between inputs and teacher representations. We analyze the predictor's role and shows that, when optimal, it enables approximating $H(Z_\theta|(Z_\phi, X))$. We explain how the asymmetric update maximizes this bound, and then derive incremental convergence dynamics of teacher entropy and alignment, which enables a better understanding of the role played by the different information-theoretic quantities at play. Eventually, we propose a simple MI-based regularizer on the student latent space that ensures monotonic MI growth and improves performance on several natural and medical datasets. Our main contributions are:

1. We clarify the role of the predictor, showing that, when optimal, it enables approximating the quantity $H(Z_\theta \mid (Z_\phi, X))$.

2. We provide a novel information-theoretic analysis showing that predictor, stop-gradient, and EMA jointly maximize $I(Z_\theta; (Z_\phi, X))$ a lower bound on $I(Z_\theta; X)$.

3. We derive incremental convergence dynamics of teacher entropy and alignment, and show that it directly depends on the ratios student/teacher alignment and student/teacher entropy.

4. We introduce an MI-based regularization on the student latents, which improves downstream performance on both natural and medical imaging benchmarks.

## 2 BACKGROUND AND RELATED WORKS

**Multi-view self-supervised learning (MVSSL) and mutual information.** Contrastive learning (CL) builds on an intuition that dates back to early InfoMax formulations Linsker (1988); Bell & Sejnowski (1995); Becker & Hinton (1992): different views of the same input should share common information. Given a sample $x$ and two transformations $v, v^+$ producing augmented views, an encoder $f_\theta$ is trained so that their embeddings $f_\theta(v)$ and $f_\theta(v^+)$ preserve this shared information. This is typically formalized by maximizing their agreement, often through mutual information, i.e., $\theta^* = \arg\max I(f_\theta(v); f_\theta(v^+))$. As noted by Tschannen et al. (2019), this objective can be interpreted as a lower-dimensional surrogate of the classical InfoMax principle $\max_\theta I(x; f_\theta(x))$, with the choice of augmentations acting as an inductive bias on what information is retained. In practice, modern MVSSL instantiates this idea through the InfoNCE loss Oord et al. (2019); Poole et al. (2019); Tian et al. (2020), which encourages agreement between positive pairs while repelling negatives, and provides a lower bound on the MI between view representations.

InfoNCE requires many negatives to estimate MI effectively Poole et al. (2019), motivating large batch sizes. MoCo He et al. (2020); Chen et al. (2020b) addressed this by introducing a large queue of negatives, while Decoupled Contrastive Learning (DCL) Yeh et al. (2022) reduced sensitivity to batch size by removing the negative–positive coupling term. Wang & Isola (2020) further analyzed CL objectives through the lens of alignment and uniformity, showing that the negative term promotes entropy maximization on the hypersphere. Extending this, Louiset et al. (2024) used a kernel density estimation (KDE) analysis to formalize the link between alignment, uniformity, and MI, explaining why large batches are often necessary. They showed that, under i.i.d. assumptions, DCL directly maximizes $I(X; Z)$, with alignment minimizing conditional entropy $H(Z|X)$ and repulsion maximizing marginal entropy $H(Z)$. Moreover, they established that the uniformity measure of Wang & Isola (2020) is a Jensen lower bound of $H(Z)$ and, when combined with alignment, provides a bound on $I(X; Z)$. Due to its pairwise nature, uniformity is thus computationally less expensive than entropy estimation and converges to the same minima as maximizing $H(Z)$, making it a reliable surrogate for entropy in both analysis and practice.

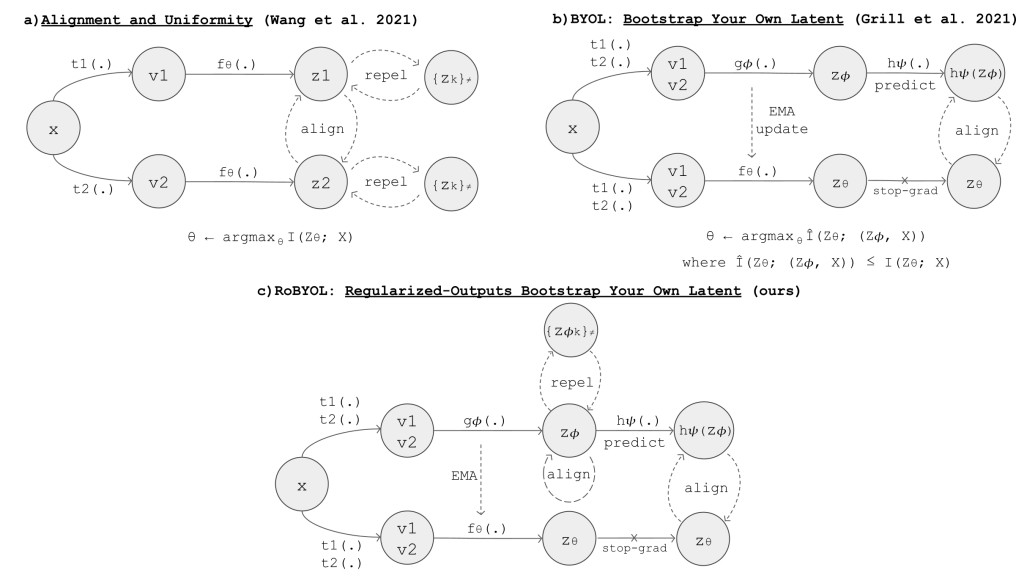

Figure 1: **Overview of the connection between self-supervised learning and mutual information.** (a) Alignment and Uniformity (Wang et al., 2021) learns representations by aligning positive pairs while promoting uniform coverage of the latent space through negative pair repulsion. (b) BYOL (Grill et al., 2021) avoids negative samples by maintaining an online student encoder and a target teacher encoder. The teacher is updated via EMA, while the student learns by predicting the teacher embedding of a different view (e.g., predicting $v^2$ teacher embedding from $v^1$ student embedding) through a prediction head. (c) Regularized-Outputs BYOL (RoBYOL, ours) extends BYOL by introducing a mutual information–based regularization on the student's outputs.

**Clustering and variance–invariance methods.** Clustering-based MVSSL methods (*e.g.*, Deep-Cluster, SwAV) promote invariance by assigning augmented views of the same input to a shared prototype. This encourages high-variance, structured embeddings and can be interpreted as maximizing MI between views and discrete prototype assignments Caron et al. (2018; 2020); Rodríguez-Gálvez et al. (2023). Variance–invariance methods (*e.g.*, Barlow Twins, VICReg) remove the need for negatives by enforcing invariance while preserving variance and reducing redundancy across features. Information-theoretic analyses show that it optimizes MI-like objectives and controls representations redundancy Zbontar et al. (2021); Bardes et al. (2021); Shwartz-Ziv et al. (2023).

**Teacher–student methods and their convergence.** Non-contrastive teacher–student methods such as BYOL Grill et al. (2020) and SimSiam Chen & He (2021) achieve strong performance without negatives, clustering, or explicit variance regularization. They rely on an asymmetric architecture with a predictor and a stop-gradient operation to avoid collapse, and in BYOL, an exponential moving average (EMA) teacher update. BYOL conjectured that its updates destabilize collapsed solutions by increasing variance, while SimSiam interpreted its alternating optimization as an Expectation–Maximization–like procedure over latent variables, with the stop-gradient enforcing a two-step update. The precise role of the predictor and EMA, however, remains poorly understood.

Several theoretical studies analyze this stability. Tian et al. (2021) showed that predictor, stop-gradient, and weight decay are jointly necessary to prevent collapse in linear settings, with the predictor aligning its eigenspace with the feature correlation matrix. Richemond et al. (2021) proved that a linear predictor converges to the identity, while Halvagal et al. (2023) used NTK analysis to show eigenvalues of the student covariance are driven toward one, equalizing variance. Wang et al. (2022) demonstrated that weight decay suppresses augmentation-variant directions, implicitly enforcing invariance. From a spectral perspective, Zhuo et al. (2023) argued that the predictor acts as a low-pass filter, ensuring the teacher has higher effective rank and driving the student toward richer features. In general, the exact role of the predictor in non-linear networks remains poorly understood, and the interest brought by the two-step optimization process remains vague.

**Mutual Information and Masked Autoencoders.** Another family of methods achieving strong performance in image representation learning is masked autoencoders (MAEs). These approaches use encoder–decoder Vision Transformers (ViTs) to reconstruct the original input from a heavily masked image, encouraging the latent space to capture informative and generalizable features. Recent studies have analyzed MAEs through the lens of contrastive learning Kong & Zhang (2023); Huang et al. (2024); Zhang et al. (2022b), and Huang et al. (2025) framed them under the information bottleneck principle, balancing relevant and irrelevant information in the latent space to improve downstream performance. In Appendix, Sec. B we provide a novel theoretical demonstration showing that MAEs can be interpreted as maximizing a lower bound on the Mutual Information $I(Z; X)$ under a reconstruction constraint. This connects MAEs to the InfoMax principle and highlights an unexplored theoretical link between reconstruction-based and contrastive approaches.

## 3 TS-SSL METHODS IMPLICITLY MAXIMIZE AN INFOMAX LOWER BOUND

In this section we show that the two-step teacher–student optimization used in BYOL and SimSiam implicitly maximizes a lower bound of the MI between inputs $X$ and the teacher representations $Z_\theta$.

**Background and Notations.** Let $\mathcal{X} = \{x_i\}_{i=1}^N$ denote the input dataset with $x_i \sim p_X$. Self-supervised learning operates on multiple stochastic augmentations of the same input. To make this dependence structure explicit, we sample two augmentations independently as $t, t' \overset{\text{iid}}{\sim} \mathcal{T}$ and define the resulting views as $v = t(x)$ and $v' = t'(x)$, where $v, v' \in \mathcal{V}$ denote images in the augmented view space. We introduce a teacher encoder $f_\theta : \mathcal{V} \to \mathbb{R}^d$ that maps each view to a latent representation, i.e., $z_\theta = f_\theta(v)$ for a specific view $v$. The randomness induced by $X$ and the augmentation process defines the random variable $Z_\theta := f_\theta(t(X))$, where $t(.)$ is an augmentation randomly drawn for each image. Similarly, the student encoder $f_\phi : \mathcal{V} \to \mathbb{R}^d$ produces $z_\phi = f_\phi(v')$ and the associated random variable $Z_\phi := f_\phi(t'(X))$, where again, $t(.)$ is an augmentation randomly drawn for each image, and $t$ and $t'$ are also independently drawn. This matches the BYOL/SimSiam setup, where the teacher and student process *different views of the same input*. A predictor $h_\psi : \mathbb{R}^d \to \mathbb{R}^d$ further transforms the student representation to align with the teacher space, producing $z_{\phi,\psi} = h_\psi(z_\phi)$ and $Z_{\phi,\psi} := h_\psi \circ f_\phi(t'(X))$. Lowercase symbols denote single-view representations, while uppercase symbols denote the corresponding random variables, ensuring that mutual information is always defined between random variables rather than specific realizations.

**InfoMax Principle.** Our goal is to learn the parameters $\theta$ that maximize the mutual information (MI) between the inputs $X$ and the teacher representations $Z_\theta = f_\theta(t(X))$:

$$I(X; Z_\theta) = -\mathbb{E}_{x \sim p_X} H(Z_\theta \mid X = x) + H(Z_\theta), \tag{1}$$

where the first term enforces alignment of representations across augmentations and the second encourages high-entropy (uniform) representations Louiset et al. (2024) Wang & Isola (2020).

**BYOL and SimSiam's Loss.** To be explicit about the empirical loss optimized in practice: BYOL and SimSiam trains the student-predictor pair to minimize a prediction loss of the teacher representation, typically of the form

$$\mathcal{L}_{\text{pred}}(\theta, \phi, \psi) = \mathbb{E}_{x \sim \mathcal{D}}\Big[\ell\Big(\text{stop-grad}(f_\theta(t(x))), \ h_\psi(g_\phi(t'(x)))\Big)\Big], \tag{2}$$

where $\ell(\cdot, \cdot)$ is a similarity-based loss (e.g., mean squared error after $\ell_2$-normalization or negative cosine similarity). In the next paragraphs, we show that this empirical loss naturally emerges from the InfoMax principle.

**Teacher-Student Lower Bound.** We now derive a lower bound on the alignment term in the InfoMax objective by introducing the student–predictor pair $(f_\phi, h_\psi)$, which infers the teacher representation $Z_\theta$ from the student representation $Z_\phi$. We define the student–predictor conditional distribution over the teacher representation as: $q_{\phi,\psi}(z_\theta \mid X) := \hat{q}_\psi(z_\theta \mid Z_\phi = f_\phi(t'(X)))$ where $\hat{q}_\psi$ is a parametric distribution (e.g., Gaussian) centered at $h_\psi(Z_\phi)$ with fixed covariance. This makes $q_{\phi,\psi}(z_\theta \mid X)$ a tractable approximation of the teacher posterior $p_\theta(z_\theta \mid X)$.

Using the standard cross-entropy decomposition of conditional entropy (also known as the Barber & Agakov bound Poole et al. (2019)), we can write:

$$\underbrace{-\mathbb{E}_{x \sim p_X} H(Z_\theta \mid X = x)}_{\text{Alignment}} = \underbrace{-\mathbb{E}_{x \sim p_X} \mathbb{E}_{z_\theta \sim p_\theta(\cdot \mid x)}\big[\log q_{\phi,\psi}(z_\theta \mid x)\big]}_{\text{Teacher–Student Cross-Prediction}} + \underbrace{\mathbb{E}_{x \sim p_X} D_{\text{KL}}\big(p_\theta(\cdot \mid x) \,\|\, q_{\phi,\psi}(\cdot \mid x)\big)}_{\text{Teacher–Prediction KL Divergence}}.$$

$$\tag{3}$$

Since the KL term is non-negative, dropping it yields a lower bound:

$$\underbrace{-\mathbb{E}_{x \sim p_X} H(Z_\theta \mid X = x)}_{\textbf{Alignment}} \geq \underbrace{-\mathbb{E}_{x \sim p_X} \mathbb{E}_{z_\theta \sim p_\theta(\cdot \mid x)} \big[ \log q_{\phi,\psi}(z_\theta \mid x) \big]}_{\textbf{Teacher–Student Cross-Prediction}}. \tag{4}$$

Combining this with the entropy term $H(Z_\theta)$ from the InfoMax objective, we obtain a lower bound on mutual information:

$$I(X; Z_\theta) = H(Z_\theta) - \mathbb{E}_{x \sim p_X} H(Z_\theta \mid X = x)$$
$$\geq \underbrace{-\mathbb{E}_{x \sim p_X} \mathbb{E}_{z_\theta \sim p_\theta(\cdot \mid x)} \big[ \log q_{\phi,\psi}(z_\theta \mid x) \big]}_{\textbf{Teacher–Student Cross-Prediction}} + \underbrace{H(Z_\theta)}_{\textbf{Entropy}} \tag{5}$$

The bound itself is a standard result in information theory (Barber & Agakov 2003) and is not a novel mutual information property. The novelty of our work lies in demonstrating that BYOL and SimSiam implicitly maximize this bound with a two-step optimization process.

## 3.1 RETRIEVING THE NEGATIVE COSINE SIMILARITY LOSS WITH MULTIPLE VIEWS

We now show that the Teacher–Student Cross-Prediction term naturally recovers a multiview generalization of the negative cosine similarity loss used in BYOL and SimSiam (full derivation in Appendix Sec. C). First, we use a resubstitution entropy estimator, as in Wang & Isola (2020), to draw $L$ vectors $z_{\theta_i}^{(l)} = f_\theta(v_i^{(l)})$ from $p_\theta(z \mid X = x_i)$, where $v_i$ is a randomly augmented view of $x_i$, drawn from the dataset of length $N$. Then, using a kernel density estimator (KDE) with $K$ draws, (as in Louiset et al. (2024), Sec. C) with an isotropic Gaussian kernel with Identity covariance matrix, we estimate the distribution $\hat{q}_{\phi,\psi}(z_{\theta_i} \mid x_i)$ with the following formula $\hat{q}_{\phi,\psi}(z_{\theta_i}^{(l)} \mid x_i) = -\frac{d}{2}\log(2\pi\sigma^2)\exp\Big( -\frac{\|f_\theta(v_i^{(l)}) - h_\psi(f_\phi(v_i^{(k)}))\|_2^2}{2\sigma^2}\Big)$. The TS Cross-Prediction can thus be estimated as:

$$\underbrace{-\mathbb{E}_{x \sim p_X} \mathbb{E}_{z_\theta \sim p_\theta(\cdot \mid x)} \big[ \log q_{\phi,\psi}(z_\theta \mid x) \big]}_{\textbf{Teacher–Student Cross-Prediction}} = \frac{1}{N}\sum_{i=1}^{N}\frac{1}{L}\sum_{l=1}^{L} \log \hat{q}_{\phi,\psi}(z_{\theta_i}^{(l)} \mid x_i)$$
$$= -\frac{d}{2}\log(2\pi\sigma^2) + \frac{1}{N}\sum_{i=1}^{N}\frac{1}{L}\sum_{l=1}^{L} \log\Big[ \frac{1}{K}\sum_{k=1}^{K}\exp\Big( -\frac{\|f_\theta(v_i^{(l)}) - h_\psi(f_\phi(v_i^{(k)}))\|_2^2}{2\sigma^2}\Big)\Big], \tag{6}$$

where $v_i^{(k)}, v_i^{(l)}$ are $K$ and $L$ independently augmented views of $x_i$ with augmentations independently sampled from $\mathcal{T}$. The first term, $-\frac{d}{2}\log(2\pi) - d\log\sigma$, is constant with respect to the network parameters and can be dropped during optimization. In practice, for simplicity, we set $\sigma = 1$ and, for computational efficiency, we restrict to two views ($K = L = 1$). Maximizing the Teacher–Student Cross-Prediction term with respect to $(\phi, \psi)$ is then equivalent to minimizing the mean squared error (MSE) between the (frozen) teacher and predicted student codes, as in BYOL and SimSiam.

**Remark on the unit-sphere normalization and multi-views strategy.** When the representations $f_\theta(\cdot)$ and $h_\psi(g_\phi(\cdot))$ are $\ell_2$-normalized, minimizing the MSE loss is equivalent to maximizing cosine similarity, as shown in (Sec C of Appendix).

## 3.2 OPTIMAL PREDICTOR $h_{\psi^*}$ AND APPROXIMATION OF $-\widehat{H}(Z_\theta \mid Z_\phi, X)$

We now show that when the prediction head is optimal (*i.e.*, it maximizes the log-likelihood), the Teacher–Student Cross-Prediction term approximates the conditional cross-entropy $-\mathbb{E}_x \widehat{H}(Z_\theta \mid Z_\phi, X = x)$. Let us formalize the prediction head as a Maximum Log-Likelihood Estimation (MLLE) problem. We introduce a conditional distribution $\hat{q}_\psi(z_\theta \mid z_\phi, x)$ that approximates the true posterior $p(z_\theta \mid z_\phi, x)$. We assume a Gaussian parametric form: $\hat{q}_\psi(z_\theta \mid z_\phi, x) = \mathcal{N}\big(z_\theta \mid h_\psi(z_\phi), I\big)$, where $h_\psi$ is the neural network predictor and $I$ is the identity covariance. The MLLE seeks the parameters $\psi^*$ that maximize the expected log-likelihood:

$$\psi^* = \text{argmax}_\psi \mathcal{L}(\psi), \quad \mathcal{L}(\psi) := \mathbb{E}_{x \sim p_X} \mathbb{E}_{(Z_\theta, Z_\phi) \sim q(Z_\theta, Z_\phi \mid x)} \big[ \log \hat{q}_\psi(Z_\theta \mid Z_\phi, X = x) \big]. \tag{7}$$

Plugging in the Gaussian form of $\hat{q}_\psi$, we obtain (up to a constant):

$$\mathcal{L}(\psi) = -\frac{d}{2}\log(2\pi) - \frac{1}{2}\mathbb{E}_{x \sim p_X}\mathbb{E}_{(z_\theta, z'_\phi) \sim q(z_\theta, z'_\phi \mid X = x)}\|z_\theta - h_\psi(z'_\phi)\|_2^2, \tag{8}$$

which shows that maximizing the log-likelihood is equivalent, with respect to $\psi$, to minimizing the mean squared error between teacher and predicted student representations. Using Monte–Carlo sampling over the dataset, this corresponds exactly to the BYOL/SimSiam regression loss:

$$\operatorname{argmax}_\psi \mathcal{L}(\psi) \iff \operatorname{argmin}_\psi \frac{1}{N} \sum_{i=1}^N \frac{\|f_\theta(v_i) - h_\psi(f_\phi(v_i'))\|_2^2}{2}. \tag{9}$$

Hence, the MLLE predictor $\psi^*$ coincides with the minimizer of the BYOL/SimSiam regression loss. Under this optimal predictor, the TS Cross-Prediction term provides a Monte–Carlo estimate of the conditional cross-entropy:

$$-\mathbb{E}_{x \sim p_X} H(Z_\theta \mid Z_\phi, X = x) \approx -\mathbb{E}_{x \sim p_X} \widehat{H}(Z_\theta \mid Z_\phi, X = x)$$
$$= \mathbb{E}_{x \sim p_X} \mathbb{E}_{z_\phi' \sim p_{Z_\phi \mid X = x}} \mathbb{E}_{z_\theta \sim q(z_\theta \mid Z_\phi = z_\phi', X = x)} \log \hat{q}_{\psi^*}(z_\theta \mid Z_\phi = z_\phi', X = x). \tag{10}$$

Finally, combining this with the entropy term $H(Z_\theta)$ from the InfoMax objective, we obtain a lower bound on the mutual information that is tight under an optimal predictor:

$$I(X; Z_\theta) \geq \underbrace{-\mathbb{E}_{X \sim p_X} H(Z_\theta \mid Z_{\phi, \psi^*}, X)}_{\textbf{Teacher–Student Cross-Prediction}} + \underbrace{H(Z_\theta)}_{\textbf{Entropy}}$$
$$\approx \underbrace{-\mathbb{E}_{x \sim p_X} \widehat{H}(Z_\theta \mid Z_\phi, X = x)}_{\textbf{Under optimal predictor } h_{\psi^*}} + H(Z_\theta) = I(Z_\theta; (Z_\phi, X)). \tag{11}$$

In summary, assuming the predictor is optimal, the Teacher–Student Cross-Prediction term provides a tractable Monte–Carlo approximation of the conditional entropy $H(Z_\theta \mid Z_\phi, X = x)$, and the InfoMax lower bound effectively captures $I(Z_\theta; (Z_\phi, X))$.

### 3.3 Assuming that the optimal predictor is an additive map

**Assumption 1** (Predictor-as-additive-map)**.** *There exists a deterministic map $h : \mathbb{R}^d \to \mathbb{R}^d$ and a Gaussian random noise variable $G_t$, with covariance matrix $\Sigma_t$ such that, for every training iteration $t$,*

$$Z_\theta^t = h(Z_\phi^t) + G_t, \quad G_t \sim \mathcal{N}(0, \sigma_t^2 I), \ G_t \perp Z_\phi^t. \tag{12}$$

**Interpretation.** The teacher representation $Z_\theta$ can be expressed as a function of the student representation $Z_\phi$ plus a small Gaussian residual $G_t$, i.e., $Z_\theta = h_{\psi^*}(Z_\phi) + G_t$. Small residual variance $\sigma_t$ implies $H(Z_\theta \mid Z_\phi, X) \approx 0$. This resembles the assumption in Richemond et al. (2021), but we do not require the predictor to be linear. In practice, BYOL and SimSiam are designed so that $Z_\phi$ captures nearly all information needed to reconstruct $Z_\theta$. The residual $G_t$ represents small, mostly isotropic variability, separating deterministic prediction from stochastic fluctuations. [1]

### 3.4 How the two-step asymmetric update maximizes $I(Z_\theta; (Z_\phi, X))$

The practical TS-SSL training alternates two asymmetric steps:

1. **Student/predictor update (encode & regress).** Freeze the teacher parameters $\theta$, and update $(\phi, \psi)$ to maximize the teacher-student cross-prediction. As shown in Sec. 3.2, with an optimal predictor this step decreases an empirical estimate of $H(Z_\theta \mid Z_\phi, X)$ and thus increases the lower bound in equation 24.

2. **Teacher update (EMA / hard copy).** Update the teacher by temporal smoothing (EMA) of student parameters: $f_\theta^{t+1} = \tau f_\theta^t + (1 - \tau) f_\phi^t$, where $\tau \in (0, 1)$ (typically $\tau \approx 0.99$). This step distills the improved student into the teacher encoder.

Under Assumption 1, we derive a sufficient condition to prove that $I(Z_\theta; (Z_\phi, X))$ increases:

**Lemma 1** (EMA increases mutual information under an entropy–difference condition)**.** *Let $Z_\phi^t$ be the student and $Z_\theta^t$ the teacher at iteration $t$, and $X$ the input. Under Assumption 1 (Predictor-as-additive-map), if the teacher's marginal entropy satisfies*

$$H(Z_\theta^{t+1}) - H(Z_\theta^t) \geq d \log \tau, \quad \tau \in (0.99, 1), \tag{13}$$

---

[1]See Sec. F for further discussion.

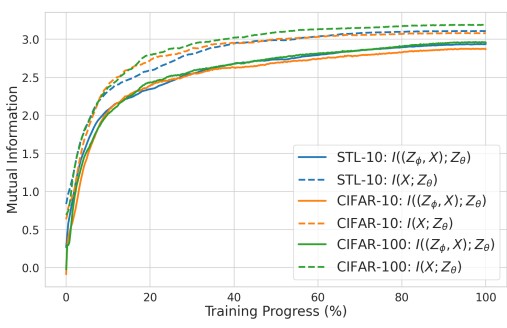 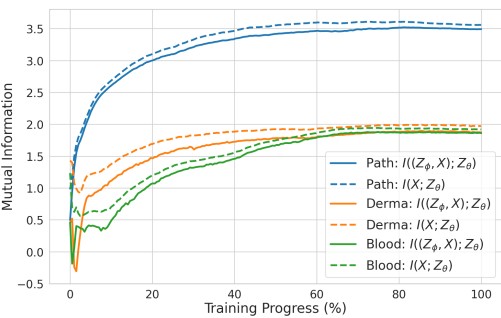

(a) Mutual information dynamics on natural datasets. (b) Mutual information dynamics on natural datasets.

Figure 2: BYOL's information-theoretic training quantities evolution: $I((Z_\phi, X); Z_\theta)$ vs $I(X; Z_\theta)$.

*then the EMA update ensures*

$$I\big((Z_\phi^t, X); Z_\theta^{t+1}\big) \geq I\big((Z_\phi^t, X); Z_\theta^t\big). \tag{14}$$

**Intuition.** The EMA update blends the teacher parameters with the student's most recent improvements, effectively transferring information the student has just extracted from $X$. This reduces the conditional uncertainty $H(Z_\theta^{t+1} \mid Z_\phi^t, X)$, making the next teacher representation more predictable from the current student. At the same time, EMA smooths parameter trajectories, preventing sudden drops in $H(Z_\theta^t)$ and ensuring that the teacher retains sufficient representational diversity. When $\tau$ is close to 1, as is standard in BYOL, where $\tau \geq 0.99$, these two effects jointly yield a monotonic increase of $I((Z_\phi^t, X); Z_\theta^{t+1})$, which is exactly the behavior predicted by Lemma 1 and observed empirically in Fig. 2. Importantly, the sufficient condition in Eq. 13 is not restrictive: for the momentum values used in practice ($\tau \geq 0.99$), we observe that it holds throughout training across six different datasets, as shown in Fig. 5 of Sec. C.2 in Appendix.

**The impact of the predictor depth.** In Supplementary Materials Sec. D.7, we empirically confirm that Lemma 1 holds for several predictor depths, with $I((Z_\phi, X); Z_\theta)$ increasing monotonically during training. However, deeper predictors yield a lower maximum mutual information, likely because they overfit teacher outputs and become overly invariant to augmentations, weakening the supervision signal for the student encoder.

## 4 AN INFORMATION-THEORETIC REGULARIZATION ON $Z_\phi$

**Assumption 2 (Linear interpolation of outputs).** We assume that the network outputs interpolate linearly under the EMA update, $\quad Z_{\theta^{t+1}} = \tau Z_{\theta^t} + (1 - \tau) Z_{\phi^t}, \quad\quad \theta^{t+1} = \tau \theta^t + (1 - \tau) \phi^t.$
*Justification.* This assumption is supported by the fact that EMA coefficients used in practice are extremely close to 1 (typically $\tau \geq 0.99$), implying that the teacher parameters move by only a very small amount per step. Since $f_\theta(x)$ is differentiable with respect to $\theta$, the teacher network can therefore be locally approximated by the first-order Taylor expansion of $f_\theta$ around the student parameters. Under this small-step regime, the EMA update in parameter space induces an approximate EMA update in output space, yielding $f_{\theta^{t+1}}(x) \approx \tau f_{\theta^t}(x) + (1 - \tau) f_{\phi^t}(x)$. Higher-order terms in the Taylor expansion scale with the square of the parameter displacement and are negligible when $\tau$ is close to 1 and when $\theta^t$ and $\phi^t$ remain close, as is typical in BYOL-style training.
Importantly, this assumption is *not required* for the mutual-information results developed in Section 3. It is only invoked in Section 4.1 to simplify the algebra and obtain an explicit, closed-form characterization of the update dynamics. All theoretical claims regarding the monotonicity of the InfoMax lower bound and the role of the EMA remain valid without this assumption. A complete derivation and discussion of approximation error are provided in Appendix, Sec. G.

### 4.1 TEACHER ALIGNMENT AND ENTROPY DYNAMICS

**Proposition (Teacher Entropy and Alignment Dynamics).** (Full details in Sec C.3 and C.4 of Appendix) Assuming that the teacher and student embeddings are jointly Gaussian and isotropic:

$$Z_{\phi^t} \sim \mathcal{N}(\mu_{\phi^t}, \sigma_{\phi^t}^2 I_d), \quad Z_{\theta^t} \sim \mathcal{N}(\mu_{\theta^t}, \sigma_{\theta^t}^2 I_d), \quad \mathrm{Cov}(Z_{\theta^t}, Z_{\phi^t}) = \rho^t \, \sigma_{\theta^t} \sigma_{\phi^t} I_d. \tag{15}$$

Assuming that with the EMA update: $Z_{\theta^{t+1}} = \tau Z_{\theta^t} + (1 - \tau) Z_{\phi^t}$, the teacher remains Gaussian and isotropic, its variance becomes: $\sigma_{\theta^{t+1}}^2 = \tau^2 \sigma_{\theta^t}^2 + (1 - \tau)^2 \sigma_{\phi^t}^2 + 2\tau(1 - \tau)\rho^t \sigma_{\theta^t} \sigma_{\phi^t}$. Hence, the incremental dynamic of the teacher entropy is:

$$H(Z_{\theta^{t+1}}) - H(Z_{\theta^t}) = \frac{d}{2} \log\left( \tau^2 + (1 - \tau)^2 \frac{\sigma_{\phi^t}^2}{\sigma_{\theta^t}^2} + 2\tau(1 - \tau)\rho^t \frac{\sigma_{\phi^t}}{\sigma_{\theta^t}} \right). \tag{16}$$

Similarly, assuming that when conditioned on an input $X_i$, the student and teacher embeddings are jointly Gaussian and isotropic with conditional variances $\sigma_{X,\theta^t}^2, \sigma_{X,\phi^t}^2$:

$$Z_{\phi^t} \mid X_i \sim \mathcal{N}(\mu_{\phi^t,i}, \sigma_{X,\phi^t}^2 I_d), \qquad Z_{\theta^t} \mid X_i \sim \mathcal{N}(\mu_{\theta^t,i}, \sigma_{X,\theta^t}^2 I_d), \quad \text{Cov}(Z_{\theta^t}, Z_{\phi^t} \mid X_i) = \rho^t \, \sigma_{X,\theta^t} \sigma_{X,\phi^t} I_d. \tag{17}$$

and correlation $\rho_X^t$, the EMA update affects the teacher alignment according to this update formula:

$$H(Z_{\theta^{t+1}} \mid X) - H(Z_{\theta^t} \mid X) = \frac{d}{2} \log\left( \tau^2 + (1 - \tau)^2 \frac{\sigma_{X,\phi^t}^2}{\sigma_{X,\theta^t}^2} + 2\tau(1 - \tau)\rho_X^t \frac{\sigma_{X,\phi^t}}{\sigma_{X,\theta^t}} \right). \tag{18}$$

**Interpretation.** The teacher entropy increases whenever the convex combination of student and teacher variances, weighted by their correlation $\rho^t$, exceeds the previous teacher variance. In particular, Eq. 15. tells us that increasing the variance ratio $\sigma_{\phi^t}^2/\sigma_{\theta^t}^2$ (respectively, Eq.16 shows that decreasing the conditional ratio $\sigma_{X,\phi^t}^2/\sigma_{X,\theta^t}^2$) promotes a higher teacher entropy (and, respectively, a better alignment with the student). Hence, variance ratios are key levers for controlling monotonic entropy growth and alignment dynamics. For SimSiam, $\tau = 0$, and the formula simplifies to $\frac{d}{2} \log \frac{\sigma_{\phi^t}^2}{\sigma_{\theta^t}^2}$ in Eq.16 and $\frac{d}{2} \log \frac{\sigma_{X,\phi^t}^2}{\sigma_{X,\theta^t}^2}$ in Eq. Eq.18.

### 4.2 REGULARIZING THE STUDENT TO INCREASE $I(Z_\theta; X)$

Our analysis indicates that the mutual information $I(Z_\theta; X)$ increases when student representations exhibit higher entropy and lower input-conditional entropy. This motivates adding a regularization term that encourages $Z_\phi$ to be both expressive and predictable from $X$. We incorporate this idea into BYOL and SimSiam, yielding *RoBYOL* and *RoSiam*, which optimize

$$\begin{cases} \phi^{t+1}, \psi^{t+1} \leftarrow \operatorname{argmin}_{\phi,\psi} \left[ - \mathbb{E}_{X \sim p_X} H\big( p_{\theta^t}(z \mid X) \,\|\, q_{\phi,\psi}(z \mid X) \big) \right], \\ \text{s.t.} \quad H(Z_\phi) \geq H(Z_\theta^t), \quad H(Z_\phi \mid X) \leq H(Z_\theta^t | X), \\ \theta^{t+1} \leftarrow \tau \theta^t + (1 - \tau)\phi^{t+1}, \quad \text{with } \tau = 0 \text{ in RoSiam.} \end{cases} \tag{19}$$

Introducing Lagrange multipliers yields the practical objective

$$\operatorname{argmin}_{\phi,\psi} \left[ \frac{1}{N} \sum_{i=1}^N \frac{\|f_\theta(v_i^{(1)}) - h_\psi(f_\phi(v_i^{(2)}))\|_2^2}{2} + \beta H(Z_\phi \mid X) - \lambda H(Z_\phi) \right], \tag{20}$$

where the first term is the standard BYOL/SimSiam loss and the entropy terms bias the student toward representations that tighten the InfoMax lower bound and promote a monotonic increase of $I(Z_\theta; X)$ during training.

In practice, we estimate $H(Z_\phi)$ via the uniformity term $U(Z_\phi)$ from Wang & Isola (2020), which is equivalent in optimum but computationally cheaper than log-sum-exp. And $H(Z_\phi | X)$ is estimated using KDE (Louiset et al., 2024) with 2 views, recovering the usual alignment term. For simplicity, and to avoid too extensive hyperparameter tuning we set $\lambda = \beta$. The resulting objective is:

$$\mathcal{L}_{\text{RoBYOL}} = \frac{1}{N} \sum_{i=1}^N \frac{\|f_\theta(v_i^1) - h_\psi(f_\phi(v_i^2))\|_2^2}{2} + \lambda \frac{1}{N} \sum_{i=1}^N \frac{\|f_\phi(v_i^1) - f_\phi(v_i^2)\|_2^2}{2}$$

$$+ \lambda \log \frac{1}{2N(N-1)} \sum_{i=1}^N \sum_{j \neq i}^N e^{-2\|f_\phi(v_i^1) - f_\phi(v_j^1)\|_2^2} + \lambda \log \frac{1}{2N(N-1)} \sum_{i=1}^N \sum_{j \neq i}^N e^{-2\|f_\phi(v_i^2) - f_\phi(v_j^2)\|_2^2}.$$

$$\tag{21}$$

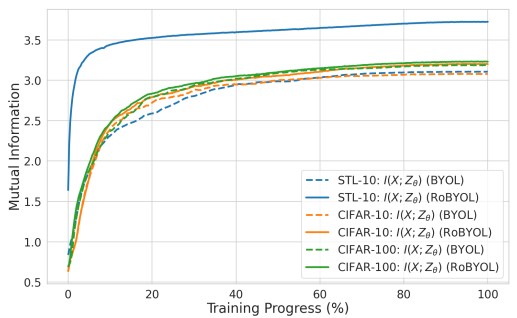 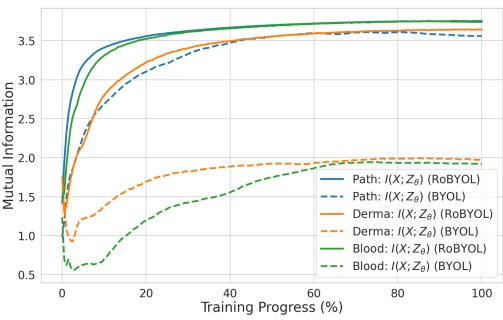

(a) Mutual information dynamics on natural datasets.   (b) Mutual information dynamics on natural datasets.

Figure 3: RoBYOL and BYOL's information-theoretic training quantities evolution: $I(X; Z_\theta)$.

We typically set $\lambda \leq 1$, since large uniformity weights can degrade performance in small-batch regimes. This encourages $H(Z_\phi) \geq H(Z_\theta)$, leading to a strict increase in the entropy of the teacher while mitigating density estimation errors. In Fig. 3, we show that RoBYOL consistently yields Mutual Information quantities larger than BYOL, where larger MI gains corresponds to larger downstream classification performance gains, as shown in the Results section (notably on STL10, BloodMNIST and DermaMNIST). Notably, we can observe that the Mutual Information $I(Z_\theta; X)$ may be upper bounded by a plateau in BYOL, while RoBYOL yields larger MI solutions.

## 5 EXPERIMENTAL RESULTS

In the following experiments, we evaluate the performance of various self-supervised learning (SSL) methods, implemented with the help of the library *solo-learn* da Costa et al. (2022), compared to our methods (*RoBYOL* and *RoSiam*) on both natural and medical imaging datasets. In all experiments, the best and second-best scores are highlighted in bold and underlined, respectively. Furthermore, we highlight in green (resp. red) the improvement (resp. the deterioration) of the proposed methods compared to the baseline methods BYOL and SimSiam. For all datasets, we study the influence of regularization strength $\lambda$ in Appendix Sec. D, while Implementation details are described in Sec. E.

**Natural imaging experiments:** we conducted experiments on four natural imaging benchmarks: CIFAR-10 (C10), CIFAR-100 (C100), STL-10 (S10) with ResNet18, and ImageNet-100 (IN100) Deng et al. (2009) with ResNet50 and ViT-Small. We compare several SSL methods, such as contrastive and variance-invariance methods, analyzing their performance on downstream classification.

**BloodMNIST, DermaMNIST and PathMNIST:** We compared several SSL methods against ours on three 2D (128x128 images) medical datasets Yang et al. (2023) BloodMNIST (8 classes, blood cell types, 11,959 train, 3,421 test images), DermaMNIST (7 classes, dermatology disease subtypes, 7,007 train, 2,005 test images) and PathMNIST (9 classes histopathological diseases, 89,996 train, 7,180 test images). In Tab. 2, we show the effectiveness of our proposed methods in improving performance on downstream classification task.

**Camelyon16 (Metastasis Detection).** We evaluate our methods on the Camelyon16 dataset Ehteshami Bejnordi et al. (2017), a benchmark for breast cancer metastasis detection from histopathology WSIs. The dataset contains 400 slides (239 normal, 160 tumor). WSIs are tiled into $256 \times 256$ patches at $10\times$ magnification, yielding $\sim 0.6M$ patches. Since only slide-level (weak) labels are available, we adopt a standard Multiple-Instance Learning (MIL) pipeline: (i) SSL-based feature extraction from patches, followed by (ii) MIL pooling for slide-level classification. We compare RoBYOL and RoSiam with ImageNet pre-training, BYOL, SimSiam, and other SSL baselines, across five MIL aggregators (MaxMIL, ABMIL Ilse et al. (2018), DSMIL Li et al. (2021), TransMIL Shao et al. (2021), and DTFDMIL Zhang et al. (2022a)). As shown in Table 4, both RoBYOL and RoSiam yield large gains over their baselines across all MIL variants, and achieve the best or second-best performance across MIL methods.

**BRACS (Breast Cancer Subtyping).** We further evaluate on the BRACS dataset, which comprises 547 WSIs labeled into seven lesion categories (N, PB, UDH, FEA, ADH, DCIS, IC) by three expert pathologists. This dataset is particularly challenging due to the inclusion of subtle atypical lesions (FEA, ADH) that are critical for early diagnosis. Following the same SSL+MIL pipeline, we report results in Table 4. RoBYOL's best performance $87.8$ outperforms BYOL's best performance $85.4$. RoSiam yields the strongest results overall with DSMIL and gained $+7.7$ points over SimSiam.

| DATASET | C10 | C100 | STL10 | STL10 | IN100 | IN100 |
|---|---|---|---|---|---|---|
| BATCH SIZE | 512 | 512 | 512 | 1024 | 512 | 512 |
| MODEL | R18 | R18 | R18 | R50 | R50 | ViT-S |
| EPOCHS | 1000 | 1000 | 200 | 400 | 400 | 400 |
| VICREG | 86.55 | 49.20 | 78.11 | 85.02 | 83.12 | 75.36 |
| SIMCLR | 90.72 | 65.43 | 83.35 | 88.43 | 79.04 | 74.24 |
| DINO | 87.22 | 62.27 | 78.83 | 84.42 | 71.84 | 64.78 |
| BARLOW TWINS | 90.45 | 68.22 | 80.63 | 84.91 | 79.63 | 74.40 |
| MOCOv2+ | 93.35 | 70.67 | 85.45 | 90.15 | 82.16 | 74.92 (v3) |
| SIMSIAM | 91.92 | 65.10 | 80.92 | COLLAPSE | 79.8 | COLLAPSE |
| RoSIAM (OURS) | 92.12 (+0.2) | 69.66 (+4.51) | 85.80 (+0.35) | 91.22 | 82.04 (+2.04) | 72.96 |
| BYOL | 93.16 | 71.53 | 82.96 | 86.66 | 84.56 | 75.72 |
| RoBYOL (OURS) | **93.39** (+0.23) | **72.14** (+0.61) | **86.00** (+3.04) | **91.37** (+4.71) | **84.72** (+0.16) | **80.84** (+5.12) |
| BYOL-MULTI-VIEWS | 93.93 | 71.65 | 79.45 | 87.28 | 83.02 | |
| RoBYOL-MULTI-VIEWS (OURS) | **95.39** (+1.46) | **71.92** (+0.27) | **86.14** (+6.69) | **92.16** (+4.92) | **83.50** (+0.48) | |

Table 1: SSL benchmark on natural datasets (C10/C100: CIFAR10/100, S10:STL10, IN100: ImageNet100). The lower part of the table compares BYOL and RoBYOL with multi-views.

| METHOD | BloodMNIST 128 | PathMNIST 128 | DermaMNIST 128 |
|---|---|---|---|
| DINO | 84.18 | 91.36 | 73.61 |
| VICREG | 97.72 | 93.03 | 74.86 |
| BARLOW TWINS | 76.56 | 92.10 | 72.92 |
| MOCOv2+ | 96.61 | 92.91 | 74.76 |
| SIMCLR | 97.83 | 92.16 | 74.46 |
| BYOL | 90.41 | 93.26 | 71.62 |
| RoBYOL (OURS) | **97.95** (+7.54) | 93.33 (+0.07) | **75.61** (+3.99) |
| SIMSIAM | 94.44 | 91.54 | 71.67 |
| RoSIAM (OURS) | 97.77 (+3.33) | **93.41** (+1.87) | 74.11 (+2.44) |

Table 2: SSL methods on BloodMNIST, DermaMNIST and PathMNIST (128x128).

| PRETRAIN. | MaxMIL | ABMIL | DSMIL | TransMIL | DTFDMIL |
|---|---|---|---|---|---|
| *ImageNet* | 66.8 | 72.9 | 60.1 | 70.2 | 73.1 |
| BARLOW TWINS | 87.3 | 88.1 | 91.2 | 91.5 | 93.9 |
| VICREG | 82.9 | 77.7 | 87.4 | 89.2 | 88.8 |
| MOCOv2+ | 88.2 | 81.8 | 91.2 | 86.8 | 90.8 |
| SIMCLR | 89.4 | 86.5 | 90.3 | 92.6 | 81.7 |
| DINO | 89.0 | **88.3** | 88.6 | 91.9 | 92.0 |
| BYOL | 73.6 | 74.1 | 73.8 | 75.1 | 78.5 |
| RoBYOL (OURS) | 90.0 $_{+16.4}$ | 87.0 $_{+12.9}$ | **93.6** $_{+9.8}$ | 91.9 $_{+16.8}$ | **94.9** $_{+16.4}$ |
| SIMSIAM | 90.1 | 81.2 | 88.8 | 89.8 | 92.7 |
| RoSIAM (OURS) | **93.9** $_{+3.8}$ | **88.3** $_{+7.1}$ | 93.0 $_{+4.0}$ | **94.7** $_{+4.9}$ | 94.7 $_{+2.0}$ |

| PRETRAIN. | MaxMIL | ABMIL | DSMIL | TransMIL | DTFDMIL |
|---|---|---|---|---|---|
| *ImageNet* | 70.3 | 78.0 | 77.3 | 76.1 | 80.5 |
| BARLOW TWINS | **87.0** | 86.9 | 78.2 | 82.7 | 87.4 |
| VICREG | 80.7 | 85.0 | 84.6 | 87.0 | 86.4 |
| MOCOv2+ | 81.1 | 75.1 | 78.2 | 79.3 | 76.0 |
| SIMCLR | 76.7 | 82.0 | 82.3 | 84.8 | 78.7 |
| DINO | 79.0 | **88.5** | 83.6 | 86.5 | **88.2** |
| BYOL | 79.5 | 85.4 | 85.3 | 83.8 | 84.7 |
| RoBYOL (OURS) | 84.6 $_{4.9}$ | 87.2 $_{+1.8}$ | 83.7 $_{-1.6}$ | 83.2 $_{-0.6}$ | 87.8 $_{+3.1}$ |
| SIMSIAM | 81.0 | 82.2 | 81.2 | 76.6 | 84.4 |
| RoSIAM (OURS) | 86.0 $_{+5.0}$ | 83.4 $_{+1.2}$ | **88.9** $_{+7.7}$ | **87.8** $_{+11.2}$ | 86.4 $_{+2.0}$ |

Figure 4: SSL for histhopathology benchmark on Camelyon16 (left) and BRACS (right).

# 6 CONCLUSION

In this work, we bridged Teacher–Student self-supervised learning (TS-SSL) with an information-theoretic perspective, shedding light on the mechanisms—such as the predictor head and stop-gradient—that underlie the success of methods like BYOL and SimSiam. We demonstrate that, under an optimal predictor, linear or not, these methods implicitly maximizes a bound of the InfoMax Principle, and that this bound approximates $I(Z_\theta; (Z_\phi, X))$. Our analysis revealed the dynamics of teacher entropy and alignment, showing how mutual information between representations and inputs tends to increase during training but may saturate prematurely. Building on this insight, we introduced RoBYOL and RoSiam, two regularized variants that explicitly encourage a monotonic increase of $I(Z_\theta; X)$ by jointly promoting higher entropy and better alignment in the student representations. Empirically, these methods achieve state-of-the-art performance among instance-discrimination-free SSL approaches and consistently boost the downstream classification performance of their baselines on both natural image and medical imaging benchmarks. Our findings suggest that information-theoretic regularization is a principled way to improve representation learning and may generalize to other SSL teacher–student frameworks such as JEPAs methods Bardes et al. (2024); Assran et al. (2023).

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

## A    TEACHER–STUDENT LOWER BOUND ON THE INFOMAX PRINCIPLE

The InfoMax principle seeks parameters $\theta^*$ of the teacher encoder $f_\theta$ that maximize the mutual information between the input $X$ and the teacher representations $Z_\theta := f_\theta(t(X))$. Decomposing the mutual information into alignment and entropy terms Louiset et al. (2024) gives:

$$I(X; Z_\theta) = \underbrace{-\mathbb{E}_{x \sim p_X} H(Z_\theta \mid X = x)}_{\textbf{Alignment}} + \underbrace{H(Z_\theta)}_{\textbf{Entropy}}. \tag{22}$$

We now introduce a student–predictor distribution $q_{\phi,\psi}(z_\theta \mid X)$, obtained by encoding $X$ with the student encoder $f_\phi$ and passing through the prediction head $h_\psi$, to derive a lower bound on the alignment term:

$$\underbrace{-\mathbb{E}_{x \sim p_X} H(Z_\theta \mid X = x)}_{\textbf{Alignment}} = \mathbb{E}_{x \sim p_X} \int_{z_\theta} p_\theta(z_\theta \mid X = x) \log \frac{p_\theta(z_\theta \mid X = x) q_{\phi,\psi}(z_\theta \mid X = x)}{q_{\phi,\psi}(z_\theta \mid X = x)} \, dz_\theta$$

$$= \underbrace{-\mathbb{E}_{x \sim p_X} \mathbb{E}_{z_\theta \sim p_\theta(\cdot \mid X = x)} \big[ \log q_{\phi,\psi}(z_\theta \mid X = x) \big]}_{\textbf{Teacher–Student Cross-Prediction}}$$

$$+ \underbrace{\mathbb{E}_{x \sim p_X} D_{\mathrm{KL}}\big(p_\theta(\cdot \mid X = x) \,\|\, q_{\phi,\psi}(\cdot \mid X = x)\big)}_{\textbf{Teacher–Prediction KL Divergence}}. $$

$$\tag{23}$$

Here, the cross-entropy term $H(p_\theta(z_\theta \mid X) \| q_{\phi,\psi}(z_\theta \mid X))$ measures how well the student–predictor distribution approximates the teacher's conditional distribution. Since the KL divergence is non-negative, we obtain a tractable lower bound on the InfoMax objective:

$$I(X; Z_\theta) \geq \mathcal{L}_{\theta,\phi} = \underbrace{-\mathbb{E}_{x \sim p_X} \mathbb{E}_{z_\theta \sim p_\theta(\cdot \mid X = x)} \big[ \log q_{\phi,\psi}(z_\theta \mid X = x) \big]}_{\textbf{Teacher–Student Cross-Prediction}} + \underbrace{H(Z_\theta)}_{\textbf{Entropy}}. \tag{24}$$

This formulation shows that optimizing the student–predictor pair $(f_\phi, h_\psi)$ to match the teacher's outputs effectively maximizes a lower bound on $I(X; Z_\theta)$, linking the BYOL and SimSiam training objectives to the InfoMax principle.

## B    MASKED AUTO-ENCODER MAXIMIZES THE INFOMAX PRINCIPLE:

In this section, we show that MAEs can also be interpreted as maximizing a bound of $I(Z; X)$ under a reconstruction constraint, when assuming that $Z$ is generated from encoding of stochastically masked inputs, thereby connecting them to the InfoMax principle.

Let $\mathcal{X} = \{x_i\}_{i=1}^N$ denote the input data set $x_i \sim p_X$. Each input is independently generated from the true latent variables $\hat{\mathcal{Z}} = \{\hat{z}_i \in \mathbb{R}^D\}_{i=1}^N$. The goal is to estimate an encoder $f_\theta$ (teacher network) that infers estimated latent factors $z_\theta$ from inputs and their masked inputs $v$ (*i.e.,* $p_\theta(z|x)$). The latent codes $z_\theta$ are generated by the encoder $f_\theta(\text{mask}(x))$, parameterized by $\theta$, where $\text{mask}(x) = v$ represents the views obtained through a stochastic masking function $\text{mask} \sim \mathcal{M}$, where $\mathcal{M}$ is a random masking process. We parameterize the distribution densities $p_\theta(z|x)$ using neural networks. The encoder network $f_\theta(.)$ represents $p_\theta(z|x)$.

**The InfoMax Principle:** The InfoMax principle seeks parameters $\theta^*$ of $f_\theta$ that maximize the mutual information between $X$ and $Z_\theta := f_\theta(t(X))$. Decomposing the mutual information into a reconstruction term and a marginal entropy term, we have:

$$I(X; Z_\theta) = \underbrace{-\mathbb{E}_{z \sim p_\theta(Z)} H(X \mid Z = z)}_{\textbf{Reconstruction term}} + \underbrace{H(X)}_{\textbf{Input-Entropy Term}}. \tag{25}$$

The input-entropy term $H(X)$ is independent of $\theta$ and does not affect optimization. The reconstruction term can be written explicitly as:

$$\underbrace{-\mathbb{E}_{z \sim p_\theta(Z)} H(X \mid Z = z)}_{\textbf{Reconstruction term}} = \mathbb{E}_{X \sim p_X} \mathbb{E}_{z \sim p_\theta(Z|X)} \log p(X \mid Z = z). \tag{26}$$

Introducing an auxiliary decoder distribution $q_\phi(X \mid Z_\theta)$, we can rewrite:

$$\underbrace{-\mathbb{E}_{z \sim p_\theta(Z)} H(X \mid Z = z)}_{\text{Reconstruction term}} = \mathbb{E}_{X \sim p_X} \mathbb{E}_{z \sim p_\theta(Z|X)} \log \frac{p(X \mid Z_\theta) q_\phi(X \mid Z = z)}{q_\phi(X \mid Z = z)}. \qquad (27)$$

Re-arranging the terms gives a cross-entropy plus KL decomposition:

$$\underbrace{-\mathbb{E}_{z \sim p_\theta(Z)} H(X \mid Z = z)}_{\text{Reconstruction term}} = \underbrace{\mathbb{E}_{X \sim p_X} \mathbb{E}_{z \sim p_\theta(Z|X)} \log q_\phi(X \mid Z = z)}_{\text{Auxiliary parameterized reconstruction term}} + \underbrace{\mathbb{E}_{X \sim p_X} D_{\text{KL}}(p(X \mid Z_\theta) \,\|\, q_\phi(X \mid Z_\theta))}_{\text{KL divergence term}}.$$

$$(28)$$

Since the KL divergence is non-negative, we obtain a lower bound on the reconstruction term:

$$\underbrace{-\mathbb{E}_{z \sim p_\theta(Z)} H(X \mid Z = z)}_{\text{Reconstruction term}} \geq \underbrace{\mathbb{E}_{X \sim p_X} \mathbb{E}_{z \sim p_\theta(Z|X)} \log q_\phi(X \mid Z = z)}_{\text{Auxiliary parameterized reconstruction term}}. \qquad (29)$$

Assuming a Gaussian parametric form for the auxiliary decoder $q_\phi(X \mid Z_\theta) = \mathcal{N}(X \mid d_\phi(Z_\theta), I)$, with $d_\phi$ a neural network and identity covariance, and using Monte–Carlo sampling, we obtain:

$$\underbrace{-\mathbb{E}_{z \sim p_\theta(Z)} H(X \mid Z = z)}_{\text{Reconstruction term}} \geq \underbrace{\frac{1}{N} \sum_{i=1}^{N} \frac{\|x_i - d_\phi(f_\theta(\text{mask}(x_i)))\|_2^2}{2}}_{-\mathcal{L}_{MAE}}. \qquad (30)$$

Thus, minimizing $\mathcal{L}_{MAE}$ corresponds to maximizing a lower bound on the InfoMax objective.

## C  THEORETICAL RESULTS

### C.1  ESTIMATING THE TEACHER–STUDENT CROSS-PREDICTION TERM AND GENERALIZING BYOL'S LOSS TO MULTIPLE VIEWS

We now estimate the conditional cross-entropy between the teacher representations and the predictor outputs, denoted

$$-\mathbb{E}_{x \sim p_X} H\big(p_\theta(z \mid X = x) \,\|\, q_{\phi,\psi}(z \mid x)\big). \qquad (31)$$

To do so, we use a *resubstitution entropy estimator* Ahmad & Lin (1976), where teacher samples are obtained from independent augmentations of each input.

$$\underbrace{-\mathbb{E}_{x \sim p_X} H\big(p_\theta(z \mid x) \,\|\, q_{\phi,\psi}(z \mid x)\big)}_{\text{Teacher–Student Cross-Prediction}} = \frac{1}{N} \sum_{i=1}^{N} \frac{1}{L} \sum_{l=1}^{L} \log \hat{q}_{\phi,\psi}\big(f_\theta(v_i^{(l)}) \mid x_i\big), \qquad (32)$$

where $v_i^{(l)} \sim \mathcal{T}$ denotes the $l$-th augmented view of $x_i$.

To estimate $\hat{q}_{\phi,\psi}(z \mid x_i)$, we employ a *kernel density estimator* (KDE) $K_\tau$ built from $L$ student representations:

$$\underbrace{-\mathbb{E}_{x \sim p_X} H\big(p_\theta(z \mid x) \,\|\, q_{\phi,\psi}(z \mid x)\big)}_{\text{Teacher–Student Cross-Prediction}} = \frac{1}{N} \sum_{i=1}^{N} \frac{1}{L} \sum_{l=1}^{L} \log K_\tau\Big(f_\theta(v_i^{(l)}), \, h_\psi(f_\phi(v_i^{(k)}))\Big), \qquad (33)$$

where $h_\psi$ denotes the predictor. We choose $K_\tau$ to be an isotropic Gaussian kernel with variance $\sigma^2$, yielding

$$- \mathbb{E}_{x \sim p_X} H\big(p_\theta(z \mid x) \,\|\, q_{\phi,\psi}(z \mid x)\big) + \frac{d}{2} \log(2\pi\sigma^2)$$

$$= \frac{1}{N} \sum_{i=1}^{N} \frac{1}{L} \sum_{l=1}^{L} \log \frac{1}{K} \sum_{k=1}^{K} \exp\left( -\frac{\|f_\theta(v_i^{(l)}) - h_\psi(f_\phi(v_i^{(k)}))\|_2^2}{2\sigma^2} \right). \qquad (34)$$

Eq. 34 constitutes a **multi-view generalization of the BYOL objective**, where the alignment is computed using $K \times L$ positive augmented views instead of the standard two-view setup ($K = L = 1$).

Finally, while we used a Gaussian kernel in the above derivation, the representations are $\ell_2$-normalized in practice, i.e.,

$$\|f_\theta(\cdot)\|_2 = 1.$$

Under such normalization, the squared Euclidean distance simplifies as

$$\|f_\theta(v_i) - f_\theta(v_j)\|_2^2 = 2 - 2\, f_\theta(v_i)^\top f_\theta(v_j),$$

making the Gaussian kernel equivalent (up to scaling) to a *von Mises–Fisher kernel* with concentration parameter $\kappa = 1/\sigma^2$. Thus, the squared error and the negative cosine similarity become proportional, recovering the standard BYOL/SimSiam loss as a special case of the multi-view Teacher–Student Cross-Prediction objective.

**Remark on the connection between Gaussian distributions and von Mises-Fisher distribution**
There is a close connection between Gaussian distribution and von Mises-Fisher distribution, suggesting that any properties uncovered in a Representation Learning setting with Gaussian distributions and/or Gaussian kernels still holds when the vectors are normalized, as already discussed in Louiset et al. (2024).

In the next paragraph, we display several equations so that the reader understand this connection. It is important to note that the derivation often generally hold when a) encoders are different (assymetric setting), b) views come from different images, or c) views are from the same image.

Let us note the kernel similarity between two representations: $f_\theta(x_i)$ and $f_\theta(x_j)$ as $K(f_\theta(x_i), f_\theta(x_j))$. Assuming that we are given a Gaussian kernel with a constant standard deviation $\sigma$, this term can be estimated as:

$$K_{\text{Gaussian}}(f_\theta(x_i), f_\theta(x_j)) = \frac{1}{\sqrt{2\pi\tau}} \exp \frac{-||f_\theta(x_i) - f_\theta(x_j)||_2^2}{2\tau} \tag{35}$$

Now, we can divide the square norm into three terms:

$$K_{\text{Gaussian}}(f_\theta(x_i), f_\theta(x_j)) = \frac{1}{\sqrt{2\pi\tau}} \exp \frac{-||f_\theta(x_i)||_2^2 - 2f_\theta(x_i)^T . f_\theta(x_j) + ||f_\theta(x_j)||_2^2}{2\tau} \tag{36}$$

Let assume that $f_\theta(x_i)$ and $f_\theta(x_j)$ are unit-normed, then this estimation get simplified into:

$$K_{\text{Gaussian}}(f_\theta(x_i), f_\theta(x_j)) = \frac{1}{\sqrt{2\pi\tau}} \exp \frac{-1 + f_\theta(x_i)^T . f_\theta(x_j)}{\tau} \tag{37}$$

which can be further simplified:

$$K_{\text{Gaussian}}(f_\theta(x_i), f_\theta(x_j)) = \frac{1}{e^1\sqrt{2\pi\tau}} \exp \frac{f_\theta(x_i)^T . f_\theta(x_j)}{\tau} \tag{38}$$

Ignoring the normalization terms, we recognize the von Mises-Fisher kernel with concentration hyper-parameter $\kappa = \frac{1}{\tau}$:

$$K_{\text{vMF}} = \frac{1}{C(\kappa)} \exp \frac{f_\theta(x_i)^T . f_\theta(x_j)}{\tau}$$

### C.2 PROOF OF LEMMA 1: EXPONENTIAL MOVING AVERAGE INCREASES THE MUTUAL INFORMATION $I(Z_\theta; (Z_\phi, X))$ UNDER THE SUFFICIENT CONDITION THAT $H(Z_\theta^{t+1}) - H(Z_\theta^t) \geq d \log \tau$

*Proof.* Let $Z_\phi^t$ be the student (online) representation and $Z_\theta^t$ the teacher (EMA) representation at iteration $t$, and define the next-step teacher representation as

$$Z_\theta^{t+1} = \tau Z_\theta^t + (1 - \tau)Z_\phi^t, \quad \tau \in (0, 1).$$

Consider the change in mutual information:

$$\Delta := I\big((Z_\phi^t, X); Z_\theta^{t+1}\big) - I\big((Z_\phi^t, X); Z_\theta^t\big) = H(Z_\phi^t, X \mid Z_\theta^t) - H(Z_\phi^t, X \mid Z_\theta^{t+1}).$$

By the chain rule for conditional entropy, we can decompose:

$$H(Z_\phi^t, X \mid Z_\theta^t) - H(Z_\phi^t, X \mid Z_\theta^{t+1}) = \big[H(Z_\phi^t \mid Z_\theta^t) - H(Z_\phi^t \mid Z_\theta^{t+1})\big] + \big[H(X \mid Z_\phi^t, Z_\theta^t) - H(X \mid Z_\phi^t, Z_\theta^{t+1})\big].$$

Because $Z_\theta^{t+1}$ is a deterministic function of $(Z_\phi^t, Z_\theta^t)$, the second bracket cancels:

$$H(X \mid Z_\phi^t, Z_\theta^t) - H(X \mid Z_\phi^t, Z_\theta^{t+1}) = 0.$$

We thus have:

$$H(Z_\phi^t, X \mid Z_\theta^t) - H(Z_\phi^t, X \mid Z_\theta^{t+1}) = \big[H(Z_\phi^t \mid Z_\theta^t) - H(Z_\phi^t \mid Z_\theta^{t+1})\big].$$

Hence,

$$\Delta \geq 0 \iff I(Z_\phi^t, Z_\theta^{t+1}) - I(Z_\phi^t, Z_\theta^t) \geq 0.$$

By further decomposing,

$$\Delta \geq 0 \iff H(Z_\theta^{t+1}) - H(Z_\theta^{t+1} \mid Z_\phi^t) - H(Z_\theta^t) + H(Z_\theta^t \mid Z_\phi^t) \geq 0.$$

Therefore,

$$\Delta \geq 0 \iff H(Z_\theta^{t+1}) - H(Z_\theta^t) \geq H(Z_\theta^{t+1} \mid Z_\phi^t) - H(Z_\theta^t \mid Z_\phi^t).$$

Under Assumption 1 (Predictor-as-additive-map), the teacher can be written as

$$Z_\theta^t = h(Z_\phi^t) + G_t,$$

where $h : \mathbb{R}^d \to \mathbb{R}^d$ is deterministic and $G_t \sim \mathcal{N}(0, \sigma_t^2 I)$ is independent Gaussian noise.

We can rewrite the EMA update as

$$\begin{aligned} Z_\theta^{t+1} &= \tau h(Z_\phi^t) + (1-\tau)Z_\phi^t + \tau G_t \\ &= \underbrace{\big[\tau h(Z_\phi^t) + (1-\tau)Z_\phi^t\big]}_{h'(Z_\phi^t)} + \underbrace{\tau G_t}_{G_{t+1}}, \end{aligned} \tag{39}$$

where we have defined a new deterministic map $h'(z) := \tau h(z) + (1-\tau)z$ and a scaled noise variable $G_{t+1} := \tau G_t \sim \mathcal{N}(0, \tau^2 \sigma_t^2 I)$. Because $G_t$ is independent of $Z_\phi^t$, so is $G_{t+1}$.

Under this Gaussian noise model, the conditional entropies are

$$\begin{aligned} H(Z_\theta^t \mid Z_\phi^t) &= H(G_t) = \frac{d}{2}\log\big(2\pi e\, \sigma_t^2\big), \\ H(Z_\theta^{t+1} \mid Z_\phi^t) &= H(G_{t+1}) = \frac{d}{2}\log\big(2\pi e\, \tau^2 \sigma_t^2\big) = H(G_t) + d\log\tau. \end{aligned} \tag{40}$$

Hence, the conditional-entropy difference simplifies to

$$H(Z_\theta^{t+1} \mid Z_\phi^t) - H(Z_\theta^t \mid Z_\phi^t) = d\log\tau, \tag{41}$$

which is negative for $\tau \in (0, 1)$. Therefore, a sufficient condition for $\Delta \geq 0$ is

$$H(Z_\theta^{t+1}) - H(Z_\theta^t) \geq d\log\tau, \tag{42}$$

i.e. the marginal entropy of the teacher must not decrease faster than the logarithmic shrinkage of the residual noise induced by the EMA. $\square$

**Empirical verification of the sufficient condition introduced in Lemma 1 (isotropic case).** Assume the teacher marginals are isotropic Gaussians, $\Sigma_t = \sigma_t^2 I$ and $\Sigma_{t+1} = \sigma_{t+1}^2 I$. Then their entropies satisfy

$$H(Z_\theta^t) = \tfrac{d}{2}\log\big(2\pi e\, \sigma_t^2\big), \qquad H(Z_\theta^{t+1}) = \tfrac{d}{2}\log\big(2\pi e\, \sigma_{t+1}^2\big), \tag{43}$$

so the entropy difference reduces to the scalar expression

$$H(Z_\theta^{t+1}) - H(Z_\theta^t) = \tfrac{d}{2}\log\frac{\sigma_{t+1}^2}{\sigma_t^2} = d\log\frac{\sigma_{t+1}}{\sigma_t}. \tag{44}$$

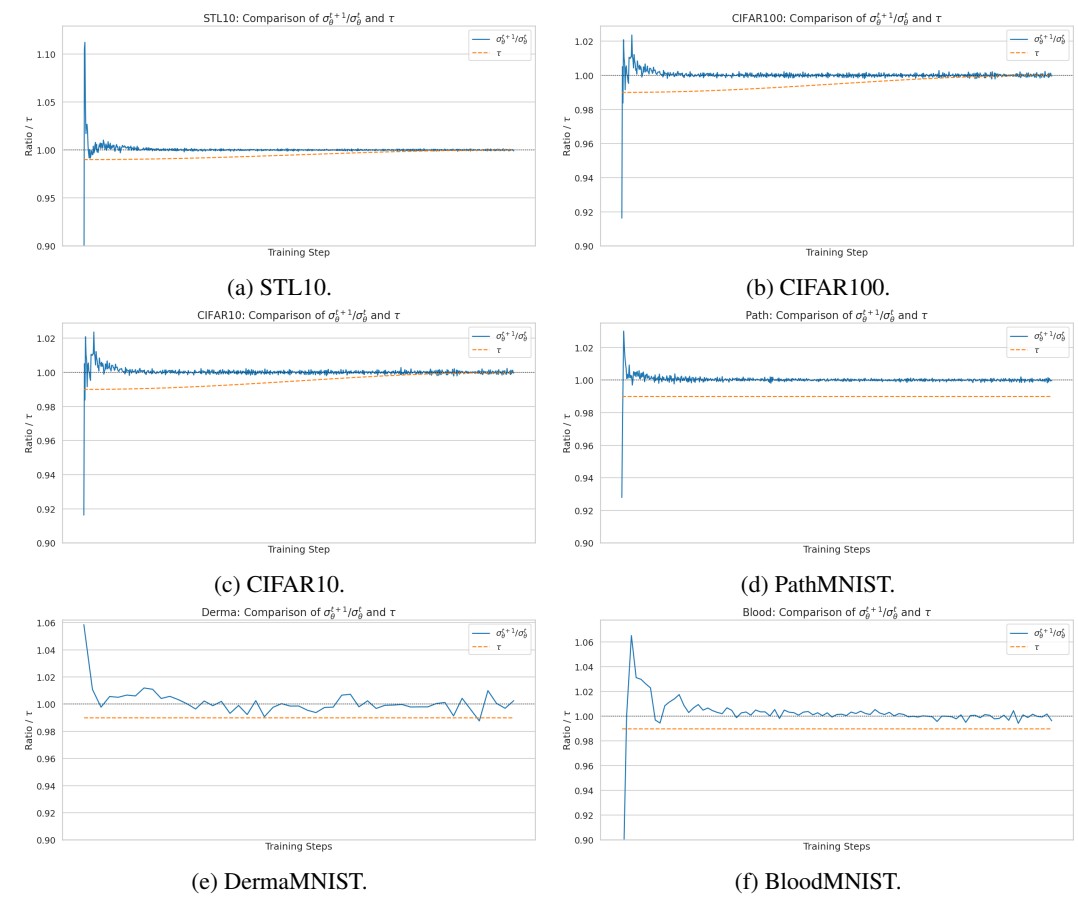

Figure 5: Empirical verification of the sufficient condition introduced in Lemma 1, in the (tractable) isotropic case on both natural and medical datasets.

Recalling the sufficient condition derived above,

$$H(Z_\theta^{t+1}) - H(Z_\theta^t) \geq d \log \tau, \tag{45}$$

we obtain the concrete isotropic check

$$\log \frac{\sigma_{t+1}}{\sigma_t} \geq \log \tau \qquad \Longleftrightarrow \qquad \frac{\sigma_{t+1}}{\sigma_t} \geq \tau. \tag{46}$$

Thus, in the isotropic regime it suffices to estimate the scalar marginal scale $\sigma_t$ (e.g. via the per-feature standard deviation of teacher activations on a held-out batch) and verify that $\sigma_{t+1} \geq \tau \sigma_t$. If inequality equation 46 holds (or approximately holds up to estimation noise), the EMA step satisfies the entropy condition required for $\Delta \geq 0$ under the Gaussian additive residual model.

In Fig 5, we show that this condition is valid during the major part of the training, except at the end, when models are stabilized anyway.

### C.3 Proof of the Teacher Alignment Dynamics formula over training

**The teacher's increase in alignment is a function of the teacher–student Pearson correlation $\rho_X^t$ and the ratio of their conditional variances.**

Assume that, conditionally on a fixed input $X_i$, the student and teacher embeddings satisfy

$$Z_{\phi^t} \mid X_i \sim \mathcal{N}(\mu_{\phi^t,i}, \sigma_{X,\phi^t}^2 I_d), \qquad Z_{\theta^t} \mid X_i \sim \mathcal{N}(\mu_{\theta^t,i}, \sigma_{X,\theta^t}^2 I_d), \tag{47}$$

and are jointly Gaussian and isotropic, with cross-covariance

$$\mathrm{Cov}(Z_{\theta^t}, Z_{\phi^t} \mid X_i) = \rho_X^t \sigma_{X,\theta^t} \sigma_{X,\phi^t} I_d, \qquad \rho_X^t \in [-1, 1]. \tag{48}$$

Let the teacher be updated by exponential moving average (EMA):

$$Z_{\theta^{t+1}} = \tau\, Z_{\theta^t} + (1 - \tau)\, Z_{\phi^t}, \qquad \tau \in (0, 1). \tag{49}$$

Define

$$A := Z_{\theta^t} - \mu_{\theta^t, i}, \qquad B := Z_{\phi^t} - \mu_{\phi^t, i}, \tag{50}$$

so that

$$\mathbb{E}[A] = \mathbb{E}[B] = 0, \quad \mathrm{Cov}(A) = \sigma_{X,\theta^t}^2 I_d, \quad \mathrm{Cov}(B) = \sigma_{X,\phi^t}^2 I_d, \quad \mathrm{Cov}(A, B) = \rho_X^t \sigma_{X,\theta^t} \sigma_{X,\phi^t} I_d. \tag{51}$$

By linearity, the centered teacher update is

$$Z_{\theta^{t+1}} - \mu_{\theta^{t+1}, i} = \tau A + (1 - \tau) B, \qquad \mu_{\theta^{t+1}, i} = \tau \mu_{\theta^t, i} + (1 - \tau) \mu_{\phi^t, i}. \tag{52}$$

Thus, the conditional covariance of the updated teacher embedding is

$$\mathrm{Cov}(Z_{\theta^{t+1}} \mid X_i) = \mathrm{Cov}(\tau A + (1 - \tau) B) \tag{53}$$

$$= \tau^2 \mathrm{Cov}(A) + (1 - \tau)^2 \mathrm{Cov}(B) + \tau(1 - \tau)(\mathrm{Cov}(A, B) + \mathrm{Cov}(B, A)) \tag{54}$$

$$= \left( \tau^2 \sigma_{X,\theta^t}^2 + (1 - \tau)^2 \sigma_{X,\phi^t}^2 + 2\tau(1 - \tau)\rho_X^t \sigma_{X,\theta^t} \sigma_{X,\phi^t} \right) I_d, \tag{55}$$

since $\mathrm{Cov}(A, B) = \mathrm{Cov}(B, A)$ in the isotropic setting.

Therefore, the updated teacher variance is

$$\sigma_{X,\theta^{t+1}}^2 = \tau^2 \sigma_{X,\theta^t}^2 + (1 - \tau)^2 \sigma_{X,\phi^t}^2 + 2\tau(1 - \tau)\rho_X^t \sigma_{X,\theta^t} \sigma_{X,\phi^t}. \tag{56}$$

The teacher's conditional entropy increase follows from the fact that an isotropic Gaussian $\mathcal{N}(\mu, \sigma^2 I_d)$ has differential entropy

$$H = \frac{d}{2} \log(2\pi e \sigma^2). \tag{57}$$

Hence,

$$H(Z_{\theta^{t+1}} \mid X) - H(Z_{\theta^t} \mid X) = \frac{d}{2} \log\left( \frac{\sigma_{X,\theta^{t+1}}^2}{\sigma_{X,\theta^t}^2} \right) \tag{58}$$

$$= \frac{d}{2} \log\left( \tau^2 + (1 - \tau)^2 \frac{\sigma_{X,\phi^t}^2}{\sigma_{X,\theta^t}^2} + 2\tau(1 - \tau)\rho_X^t \frac{\sigma_{X,\phi^t}}{\sigma_{X,\theta^t}} \right). \tag{59}$$

**Conclusion.** The alignment improves (conditional entropy decreases) whenever the conditional variance ratio $\sigma_{X,\phi^t}^2 / \sigma_{X,\theta^t}^2$ decreases, thus justifying the alignment constraint introduced in RoBYOL. $\qquad\square$

### C.4 PROOF OF THE TEACHER ENTROPY DYNAMIC FORMULA OVER TRAINING

**The teacher's increase in entropy is governed by the Pearson correlation $\rho^t$ and the variance ratio $\sigma_{\phi^t}^2 / \sigma_{\theta^t}^2$.**

Assume that the embeddings satisfy

$$Z_{\phi^t} \sim \mathcal{N}(\mu_{\phi^t}, \sigma_{\phi^t}^2 I_d), \qquad Z_{\theta^t} \sim \mathcal{N}(\mu_{\theta^t}, \sigma_{\theta^t}^2 I_d), \tag{60}$$

and are jointly Gaussian and isotropic, with cross-covariance

$$\mathrm{Cov}(Z_{\theta^t}, Z_{\phi^t}) = \rho^t \sigma_{\theta^t} \sigma_{\phi^t} I_d, \qquad \rho^t \in [-1, 1]. \tag{61}$$

Let the teacher be updated by EMA:

$$Z_{\theta^{t+1}} = \tau Z_{\theta^t} + (1 - \tau) Z_{\phi^t}, \qquad \tau \in (0, 1). \tag{62}$$

Define centered embeddings

$$A := Z_{\theta^t} - \mu_{\theta^t}, \qquad B := Z_{\phi^t} - \mu_{\phi^t}, \tag{63}$$

so that

$$\mathrm{Cov}(A) = \sigma_{\theta^t}^2 I_d, \qquad \mathrm{Cov}(B) = \sigma_{\phi^t}^2 I_d, \qquad \mathrm{Cov}(A, B) = \rho^t \sigma_{\theta^t} \sigma_{\phi^t} I_d. \tag{64}$$

The updated centered teacher embedding is

$$Z_{\theta^{t+1}} - \mu_{\theta^{t+1}} = \tau A + (1 - \tau)B, \qquad \mu_{\theta^{t+1}} = \tau \mu_{\theta^t} + (1 - \tau)\mu_{\phi^t}. \tag{65}$$

Thus,

$$\mathrm{Cov}(Z_{\theta^{t+1}}) = \left( \tau^2 \sigma_{\theta^t}^2 + (1 - \tau)^2 \sigma_{\phi^t}^2 + 2\tau(1 - \tau)\rho^t \sigma_{\theta^t} \sigma_{\phi^t} \right) I_d, \tag{66}$$

and therefore,

$$\sigma_{\theta^{t+1}}^2 = \tau^2 \sigma_{\theta^t}^2 + (1 - \tau)^2 \sigma_{\phi^t}^2 + 2\tau(1 - \tau)\rho^t \sigma_{\theta^t} \sigma_{\phi^t}. \tag{67}$$

The entropy of an isotropic Gaussian is

$$H = \frac{d}{2} \log(2\pi e \sigma^2), \tag{68}$$

so the incremental entropy update is

$$H(Z_{\theta^{t+1}}) - H(Z_{\theta^t}) = \frac{d}{2} \log\left( \frac{\sigma_{\theta^{t+1}}^2}{\sigma_{\theta^t}^2} \right) \tag{69}$$

$$= \frac{d}{2} \log\left( \tau^2 + (1 - \tau)^2 \frac{\sigma_{\phi^t}^2}{\sigma_{\theta^t}^2} + 2\tau(1 - \tau)\rho^t \frac{\sigma_{\phi^t}}{\sigma_{\theta^t}} \right). \tag{70}$$

This completes the proof. □

**Conclusion.** The teacher entropy increases whenever the variance ratio $\sigma_{\phi^t}^2/\sigma_{\theta^t}^2$ grows, and this effect is further amplified by a positive teacher–student correlation $\rho^t$. This establishes variance expansion as the principal driver of monotonic entropy growth under EMA updates, thereby justifying uniformity maximization as a mechanism for stabilizing teacher representations in TS-SSL.

# D  ADDITIONAL RESULTS

By adding our regularization term, we notably maximize the student uniformity (or entropy), thus encouraging the teacher uniformity (or entropy) to also monotonically increase. Furthermore, we also optimize student alignment. It has been previously shown Huang et al. (2023) that the generalization error bound of a downstream classifier is directly improved by a better alignment of positive pairs and a sharper concentration of augmented data. In the following experiments, we will indeed show that optimizing the mutual information $I$, and not only the uniformity (or entropy) $U$, brings to better downstream performance.

## D.1  BLOODMNIST AND PATHMNIST AT RESOLUTION 28X28

In Table 3, we report additional results on MedMNIST datasets at resolution $28 \times 28$. In Table 4, we analyze the influence of the regularization strength on downstream classification performance.

## D.2  NATURAL IMAGING EXPERIMENTS

In Table 5, we show the downstream classification performance of RoBYOL under varying regularization strengths on CIFAR10 (C10), CIFAR100 (C100), and STL10 (S10).

Table 6 shows the performance of RoSiam with varying regularization strengths.

## D.3  CAMELYON16 DATASET ABLATION STUDY

Table 7 shows the impact of different SSL pre-training methods and regularization strengths on Camelyon16 classification using various MIL strategies.

| Method | Blood 28 | Path 28 |
|---|---|---|
| Dino | 61.32 | 77.89 |
| VicReg | 90.17 | 87.78 |
| Barlow Twins | 63.31 | 83.46 |
| MoCov2+ | 88.24 | **88.68** |
| SimCLR | 91.72 | 87.55 |
| BYOL | 78.83 | 86.93 |
| RoBYOL (ours) | 92.13 (+13.20) | 88.03 (+1.10) |
| SimSiam | 85.78 | 87.64 |
| RoSiam (ours) | **92.40** (+6.82) | 87.72 (+0.08) |

Table 3: Comparison of SSL methods on BloodMNIST and PathMNIST at resolution $28 \times 28$.

| Method | Dataset | Test@1 |
|---|---|---|
| BYOL | Blood | 78.83 |
| BYOL + 0.1 $\mathcal{L}_{U(\phi)}$ | Blood 28 | 89.88 |
| BYOL + 0.5 $\mathcal{L}_{U}$ | Blood 28 | 89.79 |
| BYOL + 1 $\mathcal{L}_{U(\phi)}$ | Blood 28 | 87.95 |
| BYOL + 5 $\mathcal{L}_{U(\phi)}$ | Blood 28 | 89.12 |
| BYOL + 0.5 $\mathcal{L}_{U(Z)-H(Z|X)}$ | Blood 28 | 91.76 |
| BYOL + 1 $\mathcal{L}_{U(Z)-H(Z|X)}$ | Blood 28 | 92.13 |
| SimSiam + 0.5 $\mathcal{L}_{U(Z)-H(Z|X)}$ | Blood 28 | 91.40 |
| SimSiam + 1 $\mathcal{L}_{U(Z)-H(Z|X)}$ | Blood 28 | 92.40 |
| BYOL + 0.5 $\mathcal{L}_{U(\phi)}$ | Path 28 | 86.36 |
| BYOL + 1 $\mathcal{L}_{U(\phi)}$ | Path 28 | 86.04 |
| BYOL + 5 $\mathcal{L}_{U(\phi)}$ | Path 28 | 86.07 |
| BYOL + 5 $\mathcal{L}_{U(Z)-H(Z|X)}$ | Path 28 | 86.28 |
| BYOL + 1 $\mathcal{L}_{U(Z)-H(Z|X)}$ | Path 28 | 87.25 |
| BYOL + 0.5 $\mathcal{L}_{U(Z)-H(Z|X)}$ | Path 28 | 88.03 |
| BYOL + 0.1 $\mathcal{L}_{U(Z)-H(Z|X)}$ | Path 28 | 87.55 |
| SimSiam + 1 $\mathcal{L}_{U(Z)-H(Z|X)}$ | Path 28 | 87.61 |
| SimSiam + 0.5 $\mathcal{L}_{U(Z)-H(Z|X)}$ | Path 28 | 87.40 |

Table 4: Impact of regularization strength on downstream classification performance on MedMNIST Yang et al. (2023). Here, $U(\cdot)$ denotes uniformity and $U(Z) - H(Z|X)$ corresponds to the mutual information–based regularization.

| METHOD | C10 | C100 | STL10 |
|---|---|---|---|
| BYOL | 93.16 | 71.53 | 82.96 |
| RoBYOL (OURS) | **93.39** (+0.23) | **72.14** (+0.61) | **86.00** (+3.04) |
| RoBYOL + $\mathcal{L}_{\text{REDUNDANCY}}$ ESTEPA ET AL. (2023) | 93.06 | 71.20 | 85.58 |
| RoBYOL + $10^{-1}\mathcal{L}_{U(Z_\phi)}$ | $\times$ | $\times$ | 85.02 |
| RoBYOL + $10^{-2}\mathcal{L}_{U(Z_\phi)}$ | 93.27 | 71.06 | 86.27 |
| RoBYOL + $10^{-3}\mathcal{L}_{U(Z_\phi)}$ | 93.12 | 71.21 | 83.98 |
| RoBYOL + $10^{-4}\mathcal{L}_{U(Z_\phi)}$ | 93.27 | 71.41 | $\times$ |
| RoBYOL + $10^{-5}\mathcal{L}_{U(Z_\phi)}$ | 93.00 | 72.06 | $\times$ |
| RoBYOL + $10^{-6}\mathcal{L}_{U(Z_\phi)}$ | 93.05 | 71.65 | $\times$ |
| RoBYOL + $1\,\mathcal{L}_{U(Z)-H(Z|X)}$ | 92.2 | 69.71 | 84.76 |
| RoBYOL + $10^{-1}\,\mathcal{L}_{U(Z)-H(Z|X)}$ | 92.41 | 70.20 | 86.00 |
| RoBYOL + $10^{-2}\,\mathcal{L}_{U(Z)-H(Z|X)}$ | 92.55 | 70.62 | 85.81 |
| RoBYOL + $10^{-3}\,\mathcal{L}_{U(Z)-H(Z|X)}$ | 93.12 | 71.93 | 83.76 |
| RoBYOL + $10^{-4}\,\mathcal{L}_{U(Z)-H(Z|X)}$ | 93.39 | 72.14 | 82.95 |
| RoBYOL + $10^{-5}\,\mathcal{L}_{U(Z)-H(Z|X)}$ | 93.24 | 71.47 | $\times$ |

Table 5: Impact of regularization strength and type on downstream classification of RoBYOL (ResNet18, batch size 512) on C10, C100, and STL10. '$\times$' denotes not evaluated. Gray rows highlight regularization ablation studies.

| METHOD | C10 | C100 | STL10 |
|---|---|---|---|
| RoSiam ($\lambda = 10^{-1}$) | $\times$ | $\times$ | 85.8 |
| RoSiam ($\lambda = 10^{-2}$) | 91.87 | 68.66 | $\times$ |
| RoSiam ($\lambda = 10^{-3}$) | 92.12 | 68.70 | $\times$ |
| RoSiam ($\lambda = 10^{-3}$) | 91.79 | 67.56 | $\times$ |

Table 6: Impact of regularization strength on downstream classification for RoSiam (ResNet18, batch size 512).

## D.4 BRACS DATASET

The BRACS dataset contains 547 annotated H&E-stained WSIs across seven lesion categories. Table 9 reports classification AUCs for various SSL pre-training methods combined with MIL strategies, as well as the impact of different regularization strengths.

## D.5 CHOOSING THE STRENGTH OF THE REGULARIZATION

In Fig. 6, we observe on natural imaging benchmarks that *RoBYOL*'s performances increases for larger values of $\lambda$, namely the strength of the student MI regularization term. Nevertheless, the downstream classification performance decreases for values of $\lambda$ too large, despite the convergence of the representations toward an almost optimal entropy. This behavior has already been observed for Contrastive Learning methods in Wang & Isola (2020), and we show that it still holds in the case TS-SSL methods. This observation is crucial empirically as it enables to derive a guideline for choosing the regularization strength on natural images: one should only look for the strength $\lambda$ that maximizes the inverted U-shape curve, for values of the strength $\lambda$ between 1 and $1e-5$. For medical imaging, we can first observe that the regularization drastically enhances the performances of BYOL and RoBYOL, but choosing it is trickier, as observed on histopathological datasets.

Theoretically, having a high representational entropy (or uniformity) is desirable for at least two reasons. First, the representational entropy $H(Z_\phi)$ is an upper bound of the MI, as $I(X; Z_\phi) \leq H(Z_\phi)$, so having a high $H(Z_\phi)$ is desirable. Second, von Kügelgen et al. (2021) demonstrated that view-invariant methods with entropy (or uniformity) regularization (such as SimCLR) could

| PRETRAINING | MaxMIL | ABMIL | DSMIL | TransMIL | DTFDMIL |
|---|---|---|---|---|---|
| MOCOv3 | 87.6 | 87.0 | 88.7 | 90.2 | 89.2 |
| BYOL | 73.6 | 74.1 | 73.8 | 75.1 | 78.5 |
| RoBYOL (OURS) | 90.0$_{+16.4}$ | 87.0$_{+12.9}$ | **93.6**$_{+9.8}$ | 91.9$_{+16.8}$ | **94.9**$_{+16.4}$ |
| SimSiam | 90.1 | 81.2 | 88.8 | 89.8 | 92.7 |
| RoSiam (OURS) | **93.9**$_{+3.8}$ | **88.3**$_{+7.1}$ | 93.0$_{+4.0}$ | **94.7**$_{+4.9}$ | 94.7$_{+2.0}$ |
| RoBYOL ($\lambda = 10^{-4}$) | 86.7 | 84.7 | 85.1 | 87.6 | 85.8 |
| RoBYOL ($\lambda = 10^{-3}$) | 90.0 | 86.6 | 88.7 | 84.9 | 87.7 |
| RoBYOL ($\lambda = 10^{-2}$) | 87.9 | 85.2 | 93.6 | 88.5 | 94.9 |
| RoBYOL ($\lambda = 10^{-1}$) | 89.6 | 87.0 | 89.4 | 91.9 | 91.4 |
| RoBYOL ($\lambda = 1$) | 85.5 | 82.4 | 87.6 | 88.5 | 90.1 |
| RoSiam ($\lambda = 10^{-4}$) | 92.6 | 87.3 | 90.7 | 91.1 | 90.7 |
| RoSiam ($\lambda = 10^{-3}$) | 93.9 | 88.3 | 91.5 | 94.7 | 94.1 |
| RoSiam ($\lambda = 10^{-2}$) | 92.4 | 87.0 | 87.1 | 91.8 | 89.2 |
| RoSiam ($\lambda = 10^{-1}$) | 92.1 | 80.7 | 92.3 | 91.7 | 92.5 |
| RoSiam ($\lambda = 1$) | 92.9 | 83.7 | 93.0 | 93.8 | 94.7 |

Table 7: Instance-based SSL methods benchmark on Camelyon16 dataset at 10x. AUC scores.

| PRETRAINING | MaxMIL | ABMIL | DSMIL | TransMIL | DTFDMIL |
|---|---|---|---|---|---|
| *Regularization: Our Method vs. Uniformity Only* | | | | | |
| RoBYOL (BEST $\lambda$) | **90.0** | 87.0 | **93.6** | **91.9** | **94.9** |
| $\mathcal{L}_{\text{BYOL}} - 10^{-4}H(Z_\phi)$ | 87.0 | 74.9 | 86.5 | 86.8 | 84.2 |
| $\mathcal{L}_{\text{BYOL}} - 10^{-3}H(Z_\phi)$ | 88.9 | **89.3** | 89.3 | 91.0 | 90.3 |
| $\mathcal{L}_{\text{BYOL}} - 10^{-2}H(Z_\phi)$ | 86.5 | 86.5 | 83.5 | 88.9 | 91.1 |
| $\mathcal{L}_{\text{BYOL}} - 10^{-1}H(Z_\phi)$ | 89.2 | 83.3 | 88.3 | 88.2 | 90.5 |
| $\mathcal{L}_{\text{BYOL}} - 1H(Z_\phi)$ | 86.8 | 83.2 | 87.5 | 86.5 | 86.0 |
| RoSiam (BEST $\lambda$) | **93.9** | **88.3** | **93.0** | 94.7 | **94.7** |
| $\mathcal{L}_{\text{SimSiam}} - 10^{-4}H(Z_\phi)$ | 90.6 | 77.9 | 86.2 | 90.7 | 89.5 |
| $\mathcal{L}_{\text{SimSiam}} - 10^{-3}H(Z_\phi)$ | **93.9** | 82.9 | 92.4 | 90.0 | 89.8 |
| $\mathcal{L}_{\text{SimSiam}} - 10^{-2}H(Z_\phi)$ | 92.8 | 76.7 | 91.1 | 92.3 | 92.1 |
| $\mathcal{L}_{\text{SimSiam}} - 10^{-1}H(Z_\phi)$ | 90.8 | 78.2 | 83.0 | 94.2 | 88.1 |
| $\mathcal{L}_{\text{SimSiam}} - 1H(Z_\phi)$ | 88.0 | 84.9 | 87.7 | **95.1** | 89.2 |

Table 8: Comparison of our regularization (RoBYOL, RoSiam) vs. uniformity-only regularization on Camelyon16 (10x).

identify the invariant content partition of the representation under the condition that the entropy of the representations is maximized. This *"identifiability"* property is crucial Lehmann & Casella (2006) and desirable in SSL Roeder et al. (2021), since it implies that the representations learned from observations actually match the true underlying latent factors of generation.

| PRETRAINING | MAXMIL | ABMIL | DSMIL | TRANSMIL | DTFDMIL |
|---|---|---|---|---|---|
| BYOL | 79.5 | 85.4 | 85.3 | 83.8 | 84.7 |
| RoBYOL (OURS) | 84.6$_{4.9}$ | $\underline{87.2}_{+1.8}$ | 83.7$_{-1.6}$ | 83.2$_{-0.6}$ | $\underline{87.8}_{+3.1}$ |
| SIMSIAM | 81.0 | 82.2 | 81.2 | 76.6 | 84.4 |
| RoSiam (OURS) | $\underline{86.0}_{+5.0}$ | 83.4$_{+1.2}$ | $\underline{\mathbf{88.9}}_{+7.7}$ | $\mathbf{87.8}_{+11.2}$ | 86.4$_{+2.0}$ |
| RoBYOL ($\lambda = 10^{-4}$) | 84.2 | 84.0 | 79.0 | 81.7 | 85.5 |
| RoBYOL ($\lambda = 10^{-3}$) | 78.7 | 87.2 | 83.7 | 81.6 | 86.6 |
| RoBYOL ($\lambda = 10^{-2}$) | 84.5 | 84.1 | 79.9 | 83.2 | 87.8 |
| RoBYOL ($\lambda = 10^{-1}$) | 84.4 | 80.8 | 76.6 | 82.3 | 85.5 |
| RoBYOL ($\lambda = 1$) | 84.6 | 85.7 | 80.7 | 81.8 | 85.8 |
| RoSiam ($\lambda = 10^{-4}$) | 79.9 | 76.4 | 85.8 | 87.8 | 86.4 |
| RoSiam ($\lambda = 10^{-3}$) | 78.6 | 83.4 | 85.4 | 86.5 | 85.8 |
| RoSiam ($\lambda = 10^{-2}$) | 86.0 | 77.1 | 88.9 | 82.1 | 84.1 |
| RoSiam ($\lambda = 10^{-1}$) | 77.2 | 78.6 | 86.5 | 83.4 | 85.4 |
| RoSiam ($\lambda = 1$) | 75.0 | 83.0 | 84.3 | 78.3 | 82.7 |

Table 9: Instance-based SSL methods benchmark on BRACS dataset at 10x. AUC scores.

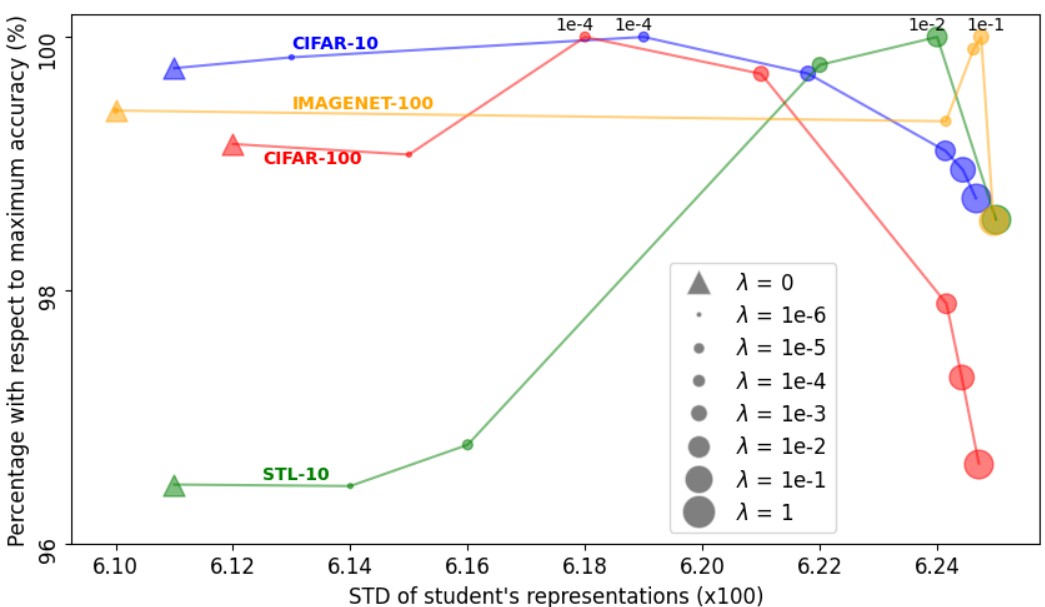

Figure 6: Impact of the regularization strength $\lambda$ on the downstream accuracy (y-axis) and entropy (x-axis) in *RoBYOL*.

### D.6 RoBYOL's Mutual Information dynamic

In Fig. 7, we analyze the dynamics of mutual information throughout training for RoBYOL compared to BYOL. Our results show that RoBYOL yields a **strictly** monotonic increase in both $I((Z_\phi, X); Z_\theta)$ and $I(X; Z_\theta)$, consistently reaching higher MI values than BYOL across datasets. This validates our intuition formulated in Sec 4.2, confirming that our regularizations tightens the lower bound and drives more informative representations. Importantly, the largest improvements in mutual information correspond to the largest gains in downstream accuracy: on medical imaging datasets, RoBYOL outperforms BYOL by +7.54% on BloodMNIST, +0.07% on PathMNIST,

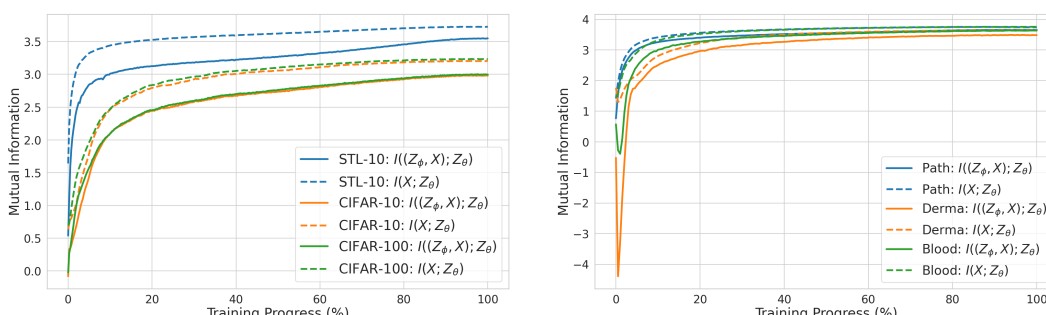

(a) Mutual information dynamics on natural datasets.  (b) Mutual information dynamics on natural datasets.

Figure 7: RoBYOL's Mutual Information evolution: $I((Z_\phi, X); Z_\theta)$ vs $I(X; Z_\theta)$.

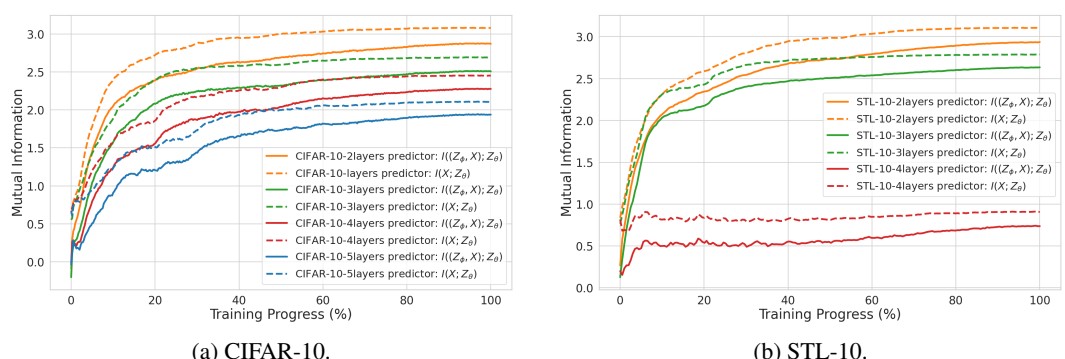

(a) CIFAR-10.  (b) STL-10.

Figure 8: **Impact of predictor depth on mutual information dynamics.** Mutual information $I((Z_\phi, X); Z_\theta)$ is shown across training epochs for predictors of increasing depth. Although the MI grows monotonically for all configurations, its asymptotic value decreases for deeper predictors, supporting the hypothesis that excessive predictor capacity can overfit the teacher outputs and weaken the encoder's ability to learn augmentation-invariant, semantically meaningful representations.

and +3.99% on DermaMNIST, highlighting that stronger information retention translates into better generalization.

## D.7 INFLUENCE OF THE PREDICTOR DEPTH

In our experiments, we further investigated the impact of predictor depth on the mutual information $I((Z_\phi, X); Z_\theta)$ during training (Fig. 8). Across both CIFAR-10 and STL-10, we observe that deeper predictors preserve the characteristic *monotonic increase* of teacher–student mutual information, in agreement with our theoretical result (Lemma 1). However, we also find that the *maximum attainable mutual information* consistently decreases as the predictor becomes deeper. We hypothesize that excessively deep predictors tend to overfit the teacher outputs and may even learn to be invariant to the stochastic augmentations applied to the input $X$. As a result, the student encoder is no longer strongly encouraged to learn augmentation-invariant features on its own, thereby reducing its ability to capture the semantic structure of the input distribution. This phenomenon highlights an important trade-off: while a certain predictor capacity is necessary for stable alignment and to avoid representational collapse, overly expressive predictors can effectively "absorb" the learning signal, weakening the training pressure on the encoder and thus lowering the quality of the learned representations.

# E   IMPLEMENTATION DETAILS

Code is entirely based on the library solo-learn da Costa et al. (2022). The alignment and uniformity regularizations terms follow the implementation of `https://www.tongzhouwang.info/hypersphere/`. Full training codes will be made available upon acceptance.

## E.1   CIFAR10 AND 100

On CIFAR10 and 100, we train SSL methods during 1000 epochs. Images are size 28x28. For each SSL methods benchmarked, we use the hyperparameters described in the library solo-learn da Costa et al. (2022), at the sole difference that we multiply the batch size (and learning rates) by 2, which makes a batch size of $512$. The best performing setup for *RoBYOL* on CIFAR10 and 100 was with $\lambda$ equals to $1e - 4$. The data augmentations used for *RoBYOL* and *RoSiam* was with two views, with RandomResizedCrop of size 32, (0.08, 1). A color jitter with prob=0.8, brightness: 0.4, contrast: 0.4, saturation: 0.2, hue: 0.1; grayscale:, prob: 0.2, and horizontal flip with prob: 0.5. We used a ResNet18 encoder with a 2 layers MLP for the projection head, with hidden dim of 4096 and output dim of 256. And a 2 layers prediction head, with a hidden dim of 4096.

For all experiments, we trained self-supervised learning models on images at a resolution of $28 \times 28$ pixels using a ResNet-18 backbone. We compared several state-of-the-art SSL methods: **BYOL**, **DINO**, **SimCLR**, **SimSiam**, **MoCo v2+**, **VICReg**, and **Barlow Twins**. For RoBYOL and BYOL, we used a projection head with hidden dimension 4096 and output dimension 256, a predictor with hidden dimension 4096, and an evolutive EMA momentum coefficient from $\tau = 0.99$ to $\tau = 0.999$. RoBYOL's regularization strength was chosen as 0.0001 for both datasets. Training was performed with LARS (batch size 512, learning rate 2.0, weight decay $10^{-5}$), using a warmup-cosine learning rate scheduler for 200 epochs. For DINO, we used a projection head with hidden dimension 2048, output dimension 256, and 4096 prototypes, with a momentum schedule from $\tau = 0.9995$ to $\tau = 1.0$, and AdamW optimizer (learning rate $5 \times 10^{-4}$, weight decay 0.04). SimCLR was trained with a temperature of 0.2, projection head of $2048 \rightarrow 256$, LARS optimizer (batch size 256, learning rate 0.4), and 200 epochs. RoSiam and SimSiam used the same backbone and projection head but with a 512-dimensional predictor and SGD optimizer (learning rate 0.5, weight decay $10^{-5}$), trained for 400 epochs. MoCo v2+ employed a momentum encoder ($\tau = 0.99 \rightarrow 0.999$), queue size of $32,768$ for both datasets, projection head of $2048 \rightarrow 256$, temperature 0.2, and SGD optimizer (batch size 512, learning rate 0.3). VICReg used a 2048-dimensional projection head and balanced the variance, invariance, and covariance losses with weights $25 : 25 : 1$, trained with LARS (batch size 512, learning rate 0.3). Barlow Twins used a projection head of $2048 \rightarrow 2048$ with cross-correlation loss scaled by 0.1. All models were trained with mixed precision (FP16), and warmup-cosine learning rate schedules.

## E.2   STL10

On STL10, we train SSL methods during 200 and 400 epochs. Images are size 96x96. For each SSL methods benchmarked, we use the hyperparameters described in the library solo-learn da Costa et al. (2022), at the sole difference that we multiply the batch size (and learning rates by two) by 2 and 4, which makes a batch size of $512$ and $1024$. The best performing setup for *RoBYOL* and *RoSiam* was with a $\lambda$ equals to $1e - 1$. The data augmentations used for *RoBYOL* and *RoSiam* was with two views, with RandomResizedCrop of size 96, (0.08, 1). A color jitter with prob=0.8, brightness: 0.4, contrast: 0.4, saturation: 0.2, hue: 0.1; grayscale:, prob: 0.2, and horizontal flip with prob: 0.5. One of the crop had no solarization, while the other had default solarization with probability 0.2. We used a ResNet18 or ResNet50 encoder with a 2 layers MLP for the projection head, with hidden dim of 4096 and output dim of 256. And a 2 layers prediction head, with a hidden dim of 4096.

We trained several self-supervised learning methods on STL-10 using asymmetric augmentations. For **Barlow Twins**, we used a $2048 \rightarrow 2048$ projection head with cross-correlation loss scaled by 0.1, optimized with LARS (learning rate 0.3). **BYOL** used a projection head of $4096 \rightarrow 256$ and a 4096-dimensional predictor, with an exponential moving average momentum $\tau \in [0.99, 1.0]$, and was optimized with LARS (learning rate 2.0, weight decay $10^{-5}$). **DINO** used a projection head $2048 \rightarrow 256$, 1024 prototypes, teacher temperature 0.04, and a momentum schedule $\tau = 0.996 \rightarrow 1.0$, trained with AdamW (learning rate $5 \times 10^{-4}$, weight decay 0.04). **MoCo v2+** employed a projection head $2048 \rightarrow 256$, queue size 32768, temperature 0.2, and momentum

$\tau = 0.99 \to 0.999$, optimized with SGD (learning rate 0.6). **SimCLR** was trained with temperature 0.2 and LARS (learning rate 0.8). **SimSiam** used a $2048 \to 2048$ projection head and a 512-dimensional predictor, with SGD (learning rate 1.0, weight decay $10^{-5}$). Finally, **VICReg** used a $2048 \to 2048$ projection head and balanced the invariance, variance, and covariance losses with weights $25 : 25 : 1$, optimized with LARS (learning rate 0.3). All models were trained with a warmup-cosine learning rate scheduler, and, mixed-precision (FP16).

We used an asymmetric augmentation pipeline to generate two views of each image. Both views applied random resized cropping with scale uniformly sampled from $[0.08, 1.0]$, random horizontal flipping with probability 0.5, color jittering with probability 0.8 (brightness and contrast 0.4, saturation 0.2, hue 0.1), and random grayscale conversion with probability 0.2. The first view did not use solarization, while the second view additionally applied random solarization with probability 0.2. Each crop was resized to $96 \times 96$ pixels, resulting in two distinct crops per sample.

### E.3 IMAGENET100

The dataset ImageNet 100 was obtained through the following Kaggle URL: `https://www.kaggle.com/datasets/ambityga/imagenet100/data`.

On ImageNet100, we train SSL methods during 400 epochs. Images are size 224x224. For each SSL methods benchmarked, we use the hyperparameters described in the library solo-learn da Costa et al. (2022), at the sole difference that we multiply the batch size (and learning rates) by 2, $128 \to 256$, distributed on two GPUs A100, which makes an effective batch size of 512. The best performing setup for *RoBYOL* and *RoSiam* was with a $\lambda$ equals to $1e - 2$. The data augmentations used for *RoBYOL* and *RoSiam* was with two views, with RandomResizedCrop of size 224, (0.08, 1). A color jitter with prob=0.8, brightness: 0.4, contrast: 0.4, saturation: 0.2, hue: 0.1; grayscale:, prob: 0.2, and horizontal flip with prob: 0.5. One of the crop had no solarization and a default torchvision gaussian blur with probability equal to 1, while the other had default solarization with probability 0.2, and a default gaussian blur with probability 0.1. We used a ResNet50 encoder with a 2 layers MLP for the projection head, with hidden dim of 4096 and output dim of 256. And a 2 layers prediction head, with a hidden dim of 4096.

We trained multiple self-supervised learning methods on ImageNet-100 using distributed data-parallel training with synchronized batch normalization, mixed precision (FP16), and a warmup-cosine learning rate schedule. **Barlow Twins** used a $2048 \to 2048$ projection head and a loss scaling factor of 0.1, optimized with LARS (learning rate 0.6, weight decay $10^{-4}$). **BYOL** employed a $4096 \to 256$ projection head with a 4096-dimensional predictor, an exponential moving average momentum $\tau \in [0.99, 1.0]$, and LARS (learning rate 1.0, weight decay $10^{-6}$). **DINO** used a $2048 \to 256$ projection head, 4096 prototypes, teacher temperature 0.04, and momentum $\tau \in [0.9995, 1.0]$, optimized with LARS (learning rate 0.6, weight decay $10^{-6}$). **MoCo v2+** was trained with a projection head $2048 \to 256$, queue size 65536, temperature 0.2, momentum $\tau \in [0.99, 0.999]$, and SGD (learning rate 0.6, weight decay $10^{-4}$). **SimCLR** used a $4096 \to 512$ projection head, temperature 0.2, and LARS (learning rate 0.6, weight decay $10^{-4}$). **SimSiam** used a $2048 \to 2048$ projection head, 512-dimensional predictor, and SGD (learning rate 1.0, weight decay $10^{-5}$), with additional entropy regularization scaled by 0.01. Finally, **VICReg** used a $2048 \to 2048$ projection head and loss weights $25 : 25 : 1$ for invariance, variance, and covariance terms, optimized with LARS (learning rate 0.3, weight decay $10^{-4}$). We also trained ViT-small, with an optimizer AdamW, with learning rate 6.e-4, and weight decay: 0.1. Instead of Mocov2+, when using ViT-S, we train Mocov3. **MoCo v3** extends Momentum Contrast by using a symmetric loss between online and momentum encoders and adopting a predictor network, similar to BYOL. In our experiments, we trained MoCo v3 with a ViT-Small backbone, a $4096 \to 256$ projection head, and a 4096-dimensional predictor using an AdamW optimizer and a cosine learning rate schedule. Unless otherwise specified, we used an asymmetric augmentation pipeline with two crops per image. Both crops applied random resized cropping with scale uniformly sampled from $[0.08, 1.0]$, random horizontal flipping ($p = 0.5$), color jitter ($p = 0.8$, brightness and contrast 0.4, saturation 0.2, hue 0.1), and random grayscale conversion ($p = 0.2$). The first view always applied Gaussian blur ($p = 1.0$) without solarization, whereas the second view applied Gaussian blur with probability 0.1 followed by random solarization ($p = 0.2$). All crops were resized to $224 \times 224$ pixels. For Sim-CLR, a symmetric augmentation pipeline was used, with stronger color jitter (brightness, contrast, saturation 0.8, hue 0.2) and Gaussian blur with probability 0.5, producing two independent crops.

### E.4 MEDMNISTv2

We conducted our self-supervised learning experiments on three datasets from MedMNIST v2: PathMNIST, BloodMNIST, and DermaMNIST, each resampled to a resolution of 128×128 pixels. PathMNIST contains histopathology images from colon tissue slides annotated into nine classes, BloodMNIST consists of blood cell images from eight morphological categories, and DermaM-NIST includes dermatoscopic images of pigmented skin lesions across seven diagnostic classes. For PathMNIST and BloodMNIST, we applied the following augmentations: random resized cropping (minimum scale 0.2, maximum scale 1.0), color jittering (probability 0.8 with brightness and contrast set to 0.4, saturation to 0.2, and hue to 0.1), random grayscale conversion (probability 0.2), and horizontal flipping (probability 0.5), generating two crops of size 128×128 per image. Gaussian blur, solarization, and equalization were disabled. For DermaMNIST, we used a slightly stronger augmentation strategy to account for its smaller sample size: random resized cropping (minimum scale 0.4, maximum scale 1.0), random rotation (always applied with a maximum angle of 25°), color jittering (probability 0.8 with brightness, contrast, and saturation set to 0.1 and hue to 0.05), random grayscale conversion (probability 0.2), Gaussian blur (probability 0.2), and horizontal flipping (probability 0.5), again producing two crops of size 128×128 per image. These augmentation pipelines were designed to maximize invariance learning while preserving relevant morphological features in the medical images.

For all experiments, we trained self-supervised learning models on PathMNIST, BloodMNIST, and DermaMNIST at a resolution of $128 \times 128$ pixels using a ResNet-18 backbone. We compared several state-of-the-art SSL methods: **BYOL**, **DINO**, **SimCLR**, **SimSiam**, **MoCo v2+**, **VICReg**, and **Barlow Twins**. For RoBYOL and BYOL, we used a projection head with hidden dimension 4096 and output dimension 256, a predictor with hidden dimension 4096, and a fixed EMA momentum coefficient $\tau = 0.99$. RoBYOL's regularization strength was chosen as 0.5 for BloodMNIST, 1 for DermaMNIST, and 0.01 for PathMNIST. Training was performed with LARS (batch size 512, learning rate 2.0, weight decay $10^{-5}$), using a warmup-cosine learning rate scheduler for 200 epochs. For DINO, we used a projection head with hidden dimension 2048, output dimension 256, and 4096 prototypes, with a momentum schedule from $\tau = 0.9995$ to $\tau = 1.0$, and AdamW optimizer (learning rate $5 \times 10^{-4}$, weight decay 0.04). SimCLR was trained with a temperature of 0.2, projection head of $2048 \rightarrow 256$, LARS optimizer (batch size 256, learning rate 0.4), and 200 epochs. RoSiam and SimSiam used the same backbone and projection head but with a 512-dimensional predictor and SGD optimizer (learning rate 0.5, weight decay $10^{-5}$), trained for 400 epochs. MoCo v2+ employed a momentum encoder ($\tau = 0.99 \rightarrow 0.999$), queue size of 4096 (for DermaMNIST), of $32,768$ for PathMNIST and BloodMNIST, projection head of $2048 \rightarrow 256$, temperature 0.2, and SGD optimizer (batch size 512, learning rate 0.3). VICReg used a 2048-dimensional projection head and balanced the variance, invariance, and covariance losses with weights $25 : 25 : 1$, trained with LARS (batch size 512, learning rate 0.3). Barlow Twins used a projection head of $2048 \rightarrow 2048$ with cross-correlation loss scaled by 0.1. All models were trained with mixed precision (FP16), and warmup-cosine learning rate schedules.

### E.5 CAMELYON16

**Camelyon16** is a 2-class dataset for detecting metastases in breast cancer, consisting of 400 slides: 239 normal tissue slides and 160 tumor slides. The official dataset is already split into training and test sets, but following Li et al. (2021), we further divide the training set into training and validation sets at a 9:1 ratio for hyperparameter tuning. We use CLAM's pre-processing pipeline Lu et al. (2021) to cut whole-slide images (WSIs) into 256×256 non-overlapping patches from foreground tissue regions at ×10 magnification, resulting in 0.6 million patches. Our method is compared against ImageNet pre-training as a baseline and seven different self-supervised learning (SSL) methods. We use *ResNet18* (11.7M parameters), a commonly used backbone in histopathological studies, initializing all SSL models with ImageNet-1K weights and pre-training them using the *solo-learn* library from da Costa et al. (2022) for 200 epochs, keeping SSL hyperparameters consistent with their original papers. We adopt DSMIL's code Li et al. (2021) as the foundation for our MIL training and evaluation pipeline, modifying models, optimizers, and other training components based on their official implementations. Each MIL model is trained for 100 *epochs* with a grid search to find the optimal *learning rate*, using a cosine annealing scheduler, Adam optimizer with a weight decay of 0.00001, and a batch size of one slide (*i.e.,* bag).

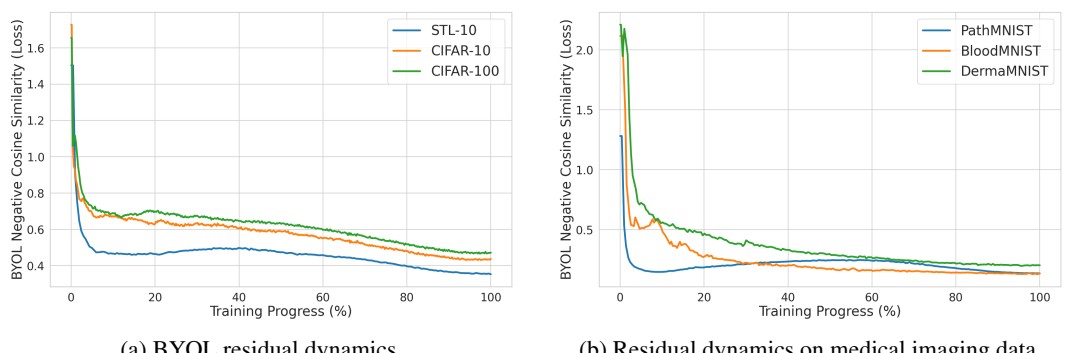

(a) BYOL residual dynamics.  (b) Residual dynamics on medical imaging data.

Figure 9: Dynamics of residuals $\|Z'_\theta - h_{\psi^*}(Z'_\phi)\|_2^2$ in the batches during training.

## F  Tracking residuals

The Fig. 9 reports the evolution of the BYOL prediction residual, measured as the negative cosine similarity between the teacher representation $Z_\theta$ and the prediction $h_\psi(Z_\phi)$, for STL-10, CIFAR-10, and CIFAR-100. Across all datasets, the residual decreases sharply during the early stages of training and then stabilizes at a substantially lower value than at initialization. This trend indicates that the predictor learns to capture most of the systematic relationship between $Z_\phi$ and $Z_\theta$, with the remaining discrepancy behaving like a relatively small stochastic component. Although the residual does not vanish completely, its monotonic decay and subsequent stabilization support the modeling assumption that the teacher representation can be approximated by a deterministic transformation of the student representation plus a noise term.

## G  Assuming that interpolating network weight is approximatively equivalent to interpolating outputs

The linear interpolation of outputs follows from the fact that the EMA coefficient $\tau$ is chosen extremely close to 1 in practice (typically $\tau \geq 0.99$). As a result, the teacher parameters change only minimally at each step, ensuring that the nonlinear network function $f_\theta(x)$ can be accurately approximated by its first-order Taylor expansion in a neighborhood of the student parameters.

**Small Step Size and Local Linearity.**   The teacher update is

$$\theta^{t+1} = \tau\theta^t + (1-\tau)\phi^t. \tag{71}$$

From the student's perspective, the teacher's parameter increment is

$$\Delta\theta' = \theta^{t+1} - \phi^t = \tau(\theta^t - \phi^t). \tag{72}$$

Since $f_\theta(x)$ is differentiable in $\theta$, we expand the new teacher output around $\phi^t$:

$$f_{\theta^{t+1}}(x) \approx f_{\phi^t}(x) + \nabla_{\phi^t} f_{\phi^t}(x)^\top (\theta^{t+1} - \phi^t). \tag{73}$$

**Relating the Gradient to Previous Outputs.**   We similarly expand the previous teacher output around $\phi^t$:

$$f_{\theta^t}(x) \approx f_{\phi^t}(x) + \nabla_{\phi^t} f_{\phi^t}(x)^\top (\theta^t - \phi^t). \tag{74}$$

Rearranging equation 74 isolates the gradient term,

$$\nabla_{\phi^t} f_{\phi^t}(x)^\top (\theta^t - \phi^t) \approx f_{\theta^t}(x) - f_{\phi^t}(x). \tag{75}$$

**Deriving the Linear Interpolation.**   Substituting $(\theta^{t+1} - \phi^t) = \tau(\theta^t - \phi^t)$ into equation 73 gives

$$f_{\theta^{t+1}}(x) \approx f_{\phi^t}(x) + \tau\left[\nabla_{\phi^t} f_{\phi^t}(x)^\top (\theta^t - \phi^t)\right], \tag{76}$$

which, using the previous identity, yields the desired interpolation rule:

$$f_{\theta^{t+1}}(x) \approx \tau f_{\theta^t}(x) + (1-\tau) f_{\phi^t}(x). \tag{77}$$

**Approximation Accuracy.** The approximation error is dominated by the second-order Taylor remainders. For the new teacher:

$$R_{\theta^{t+1}, \phi^t} = O\big(\|\theta^{t+1} - \phi^t\|^2\big) = O\big(\tau^2 \|\theta^t - \phi^t\|^2\big), \tag{78}$$

and for the previous teacher:

$$R_{\theta^t, \phi^t} = O\big(\|\theta^t - \phi^t\|^2\big) = O\big((1 - \tau)^2 \|\theta^t - \phi^t\|^2\big). \tag{79}$$

With typical EMA settings ($\tau \geq 0.99$), we have $(1 - \tau)^2 \leq 10^{-4}$, and EMA dynamics ensure that $\|\theta^t - \phi^t\|$ remains small throughout training. Thus, both second-order terms are negligible, justifying the linear interpolation of outputs.

