# OpenReview forum: "WHY TEACHER–STUDENT SELF-SUPERVISED LEARNING WORKS: A MUTUAL INFORMATION PERSPECTIVE"
_ICLR.cc/2026/Conference — Submitted to ICLR 2026_

### Official Review · Reviewer_Sd2r · 2025-10-24

**Soundness:** 1
**Presentation:** 1
**Contribution:** 2
**Rating:** 2
**Confidence:** 4

**Summary:**

The paper proposes a new multi-view self-supervised learning (SSL) loss that is motivated by the InfoMax principle, aiming to maximise a lower bound on $I(X;Z)$.

The derived loss is empirically evaluated against self-implemented baselines (Barlow Twins, VicReg, SimCLR, Dino, Barlow Twins, Mocov2, SimSiam, BYOL) on natural image datasets (Cifar10, Cifar100, STL10, Imagenet100) as well as medical image datasets (BloodMNIST, PathMNIST, DermaMNIST, Camelyon16, BRACS) and compares mostly favourably to those.

**Strengths:**

- The overview of related literature in introduction and background and related work is extensive

- It is great to see SSL methods tried out on medical imaging benchmarks

**Weaknesses:**

- The core equations of the method (Eqns 18 and 19) seem to have mistakes in them: in Eq 18 the entropy term $H(Z_{\phi})$ should be subtracted not added I believe, in Eq 19 the last term should be subtracted not added, and also the $\beta$ from Equation 18 is nowhere to be found in Eq 19. Lastly, why would the term that I think is meant to estimate $H(Z_{\phi})$ in Eq 19 (i.e. this last term) involve $f_{\theta}$?

- The natural image experiments are not conducted on full ImageNet, which has been the standard in the literature (BYOL, SimCLR, …). This raises doubts about the scalability of the proposed approach and also raises questions about fair comparison, since no independent source can be compared against for accuracy (e.g. baselines could be suboptimally trained by authors - not even on purpose. As someones who has trained these beasts before I know how hyperparameter sensitive they can be)

- Without the last term in Eq 18, the suggested modification to BYOL would look almost identical to what is proposed and investigated with an entropy regulariser in Rodriguez-Galvez et al. (2023). This warrants an ablation of its effect to better understand novelty and significance of the contribution, but it cannot be found in the paper.

**Questions:**

- Contribution 1. - what is ‘thus bringing to an estimator of […].’ supposed to mean? Bringing what?
- Eq 2, 3 and 4: how is H(p || q) to be understood? I don’t know the notation in this case, I only know || for divergence notations but not for entropy.
- Eq 2: how is $q$ defined here?
- Eq 9: how is H(A|B;C) to be understood? I would understand H(A|B) as the conditional entropy of A given B, but with a third variable separated by a semicolon I am not familiar with the notation.
- Assumption 1: how can this assumption be justified or otherwise verified in practice?
- Figure 2: how is the MI measured here?
- Where did $\beta$ from Eq 18 go in Eq 19?
- Line 478: where is this premature saturation shown in the paper?
- Line 480: typo? Should it not be $I(Z_{\phi};X)$ instead according to Eq 18 and 19?

---

> ### Author Response · Authors · 2025-12-03
>
> **Weakness 1:**
>
> **Sd2r:** "Contribution 1. - what is ‘thus bringing to an estimator of […].’ supposed to mean? Bringing what?"
>
> **Answer:**
> When stating that this "thus brings to an estimator of [...]" we mean the following: once the prediction head is optimal, it maximizes the log-likelihood of the auxiliary distribution $q_\psi(z_\theta \mid z_\phi, x)$, modeled as a Gaussian centered at $h_\psi(z_\phi)$ with unit covariance. Under this optimality, the Teacher–Student Cross-Prediction term becomes exactly the Monte–Carlo estimator of the expected log-likelihood : $
> E_x\,E_{z_\theta,z_\phi\mid x}\big[\log \hat q_{\psi^*}(z_\theta \mid z_\phi, x)\big].
> $
>
> Because $\hat q_{\psi^*}$ is Gaussian with fixed covariance, this log-likelihood is equivalent (up to an additive constant) to $-\widehat H(Z_\theta \mid Z_\phi; X)$. Therefore, having an optimal predictor directly enables the Cross-Prediction term to approximate the negative conditional entropy $-E_x\,\widehat H(Z_\theta \mid Z_\phi; X)$.
>
> This observation is, to the best of our knowledge, novel: it shows for the first time that prediction-based SSL methods such as BYOL, SimSiam, and more generally all architectures equipped with a prediction head, can be reinterpreted through the lens of conditional entropy estimation. This perspective provides a unified and principled understanding of why prediction heads are effective, and reveals that these methods implicitly maximize an information-theoretic quantity by notably estimating and reducing the conditional uncertainty $H(Z_\theta \mid Z_\phi; X)$ under an optimal predictor.
>
> ---
>
> **Weakness 2:**
>
> **Sd2r:** "Eq 2, 3 and 4: how is $H(p||q)$ to be understood? I don’t know the notation in this case, I only know $||$ for divergence notations but not for entropy. Eq 2: how is q defined here?"
>
> **Answer:**
>
> **Definition of $q_{\phi, \psi}$**
> The distribution $q_{\phi, \psi}(z\mid X)$ is the *student–predictor distribution*, obtained by first encoding $X$ through the student encoder $f_\phi$ to obtain $Z_\phi$, and then passing $Z_\phi$ through the prediction head $h_\psi$ to obtain a distribution over the same space as $Z_\theta$. Formally, we define
>
> $$
> q_{\phi, \psi}(z \mid X) := \hat q_\psi(z_\theta \mid Z_\phi = g_\phi(X)) \,,
> $$
>
> where $\hat q_\psi$ is a parametric distribution (e.g., Gaussian) centered at $h_\psi(Z_\phi)$. This makes $q_{\phi, \psi}(z_\theta \mid X)$ an approximation of the teacher distribution $p_\theta(z \mid X)$ via the student–predictor pair. We will clarify this in the main text of the manuscript.
>
> **Interpretation of $H(A \mid B; C)$**
> The notation $H(A \mid B; C)$ is a shorthand for the *conditional entropy of $A$ given $B$, under a context or conditioning variable $C$*. Concretely, in our setting
>
> $$
> H(Z_\theta \mid Z_\phi; X) := E_{x \sim p_X} \, H\big(Z_\theta \mid Z_\phi, X=x\big) \,.
> $$
>
> Here, the semicolon separates the “primary conditioning variable” $Z_\phi$ (the student representation) from the “context” $X$, emphasizing that $Z_\phi$ is derived from $X$ while $X$ itself remains part of the conditioning. We will clarify this in the main text of the manuscript.

---

> ### Author Response · Authors · 2025-12-03
>
> ### Weakness 3
>
> **Sd2r:** "Assumption 1: how can this assumption be justified or otherwise verified in practice?"
>
> **Answer:** Assumption~1 is intended as a mild and analytically convenient abstraction of the empirical behavior of prediction-based SSL methods such as BYOL and SimSiam. In these models, the teacher representation $Z_\theta$ is a smooth transformation of the student representation $Z_\phi$, and the predictor is explicitly trained to approximate this mapping. As a result, the optimal predictor $h_{\psi^*}$ captures most of the structure of the mapping $Z_\phi \mapsto Z_\theta$, while the remaining discrepancy behaves like a relatively small, approximately isotropic residual.
>
> This phenomenon can be assessed empirically by measuring the prediction error:
>
> $$
> \| Z_\theta' - h_{\psi^*}(Z_\phi) \|_2^2
> $$
>
> over the dataset. As shown in the updated manuscript (Fig.~9 of Supplementary Materials), the residual magnitude decreases rapidly during training and stabilizes at a substantially lower level than at initialization, indicating that the mapping from $Z_\phi$ to $Z_\theta$ is largely deterministic. Although the residual is not expected to be perfectly Gaussian, modeling it as additive Gaussian noise is a standard and widely used simplification in theoretical analyses: it captures the idea that the predictor explains the systematic (deterministic) component of the mapping, while the remaining variability can be treated as stochastic.
>
> Importantly, the assumption does not impose linearity or restrict the form of the predictor $h$; it merely assumes that after learning, the dominant relationship between the two representations is captured by a deterministic function, with the remaining error treated as noise. This is consistent with prior theoretical treatments of self-supervised learning and with the empirical trends observed in our experiments.
>
> To clarify we'll add the following sentences to the manuscript: "In practice, prediction-based SSL methods such as BYOL and SimSiam are explicitly designed so that $Z_\phi$ carries nearly all the information needed to reconstruct $Z_\theta$, making an additive-noise model a natural abstraction of the learned predictor dynamics. The residual term $G_t$ captures the small, largely isotropic variability that remains after prediction and provides a mathematically tractable way to separate the deterministic structure of the predictor from stochastic fluctuations. This formulation therefore offers a flexible and empirically grounded modeling assumption without imposing linearity or restricting the expressiveness of the predictor $h$.
>
> ### Weakness 4
>
> **Sd2r:** "Figure 2: how is the MI measured here?"
>
> **Answer:** The mutual information is estimated by summing the alignment (negative conditional entropy) and uniformity (which bears a close connection with entropy):
>
> $$
> I(Z_\theta; X) = U(Z_\theta) - H(Z_\theta \mid X)
> $$
>
> $$
> I(Z_\theta; (X, Z_\phi)) = U(Z_\theta) - H(Z_\theta \mid (X, Z_\phi))
> $$
>
> where $ - H(Z_\theta \mid (X, Z_\phi)) = \frac{1}{N} \sum_{i=1}^N \frac{\|f_\theta(v_i^1) - h_{\psi^*}(f_\phi(v_i^2))\|_2^2}{2}$,
>
> and $v_i^1$ and $v_i^2$ are two different views of the same image $x_i$,
>
> and $\psi^*$ is assumed to be always optimal throughout training. These values are averaged and computed within the batches.
>
> ### Weakness 5
>
> **Sd2r:** "Where did $\beta$ from Eq 18 go in Eq 19 ?"
>
> **Answer:** To simplify the parameter tuning we imposed $\beta = \lambda$. However, the reviewer is right in saying that these parameters should be conceptually different. For simplicity, we stick with $\beta = \lambda$, and we will write this clearly in the revised manuscript.
>
> ### Weakness 6
>
> **Sd2r:** "Line 480: typo? Should it not be $I(Z_\phi; X)$ instead according to Eq 18 and 19?"
>
> **Answer:** This is not a typo. According to Eq. 16 and Eq. 18 in the revised manuscript, we showed that our regularizations on $H(Z_\phi)$ help promote increasing $H(Z_\theta)$ and decreasing $H(Z_\theta \mid X)$. We will add a sentence in the manuscript to state this more clearly.
>
> ### Weakness 7
>
> **Sd2r:** "The core equations of the method (Eqs. 18 and 19) seem to have mistakes in them: in Eq. 18, the entropy term $H(Z_\phi)$ should be subtracted, not added. In Eq. 19, the last term should also be subtracted, not added, and the $\beta$ from Equation 18 is nowhere to be found in Eq. 19. Lastly, it is unclear why the term that appears to estimate $H(Z_\phi)$ in Eq. 19 (i.e., this last term) involves $f_\theta$."
>
> **Answer:** We thank the reviewer for pointing out these issues. The equations have been corrected to accurately reflect the alignment and uniformity terms on $Z_\phi$, consistent with the loss functions actually used in our experiments. We apologize for the oversight and have ensured that the revised equations now correctly represent the intended formulation.

---

> ### Author Response · Authors · 2025-12-03
>
> ### Weakness 8
>
> **Sd2r:** "Without the last term in Eq 18, the suggested modification to BYOL would look almost identical to what is proposed and investigated with an entropy regulariser in Rodriguez-Galvez et al. (2023). This warrants an ablation of its effect to better understand novelty and significance of the contribution, but it cannot be found in the paper."
>
> **Answer:** We provide extensive empirical comparisons between our MI-based regularization and uniformity-only counterparts in Tab. 8 (Sec. D.3), Tab. 5 (Sec. D.2), and Tab. 4 (Sec. D.1). Across a wide range of regularization strengths, the best-performing RoBYOL and RoSiam models generally outperform the best BYOL - $U(Z_\phi)$ variants, particularly on medical imaging benchmarks. This supports both the validity and the theoretical relevance of our approach.
>
> ### Weakness 9
>
> **Sd2r:** "Line 478: where is this premature saturation shown in the paper?"
>
> **Answer:** We apologize for not making this result clear to the readers. We revised the manuscript to include Fig. 2, where we show that RoBYOL consistently yields Mutual Information quantities larger than BYOL, where larger MI gains correspond to larger downstream classification performance gains, as shown in the Results section (notably on STL10, BloodMNIST, and DermaMNIST). Notably, we can observe that the Mutual Information $I(Z_\theta; X)$ may be upper bounded by a plateau in BYOL, while RoBYOL generally tends toward larger MI solutions.

---

### Official Review · Reviewer_zrcp · 2025-10-30

**Soundness:** 2
**Presentation:** 2
**Contribution:** 2
**Rating:** 2
**Confidence:** 4

**Summary:**

This paper investigates teacher-student (TS) style self-supervised learning and aims to provide an information-theoretic explanation for its effectiveness. Initially, the authors analyze BYOL from an information-theoretic perspective and demonstrate that the student update implicitly maximizes a lower bound on the mutual information between the representation $Z_\theta$ and the input X. Subsequently, under the assumption of a Gaussian isotropic latent space, they derive the incremental dynamics of the teacher's entropy and find that increasing the variance ratio can promote better alignment between the teacher and student. Based on this insight, the paper introduces an additional regularization term into BYOL's optimization objective. Experimental results show that this regularization improves performance.

**Strengths:**

The paper presents interesting research with adequate theoretical analysis.

**Weaknesses:**

1. Insufficient discussion of related work: The paper claims several key contributions, including theoretical analyses of the predictor, stop-gradient mechanism, and EMA, explanations for the non-collapsing behavior of TS-SSL, and the introduction of mutual information-based constraints. However, similar arguments have been made in existing literature [1, 2, 3], and the paper fails to explicitly discuss distinctions from these prior works.


2. Overstated title: TS-SSL encompasses diverse methods, such as the classic DINO. Yet, this paper only analyzes BYOL and SimSiam, which are insufficient to represent the broader TS-SSL framework, making the title overly broad.


3. Limited experimental validity: Experiments are conducted on small-scale datasets, which lack representativeness in the current era. This raises concerns about the method’s effectiveness in real-world application scenarios.

**Questions:**

In Line 344, the assumption $Z_{\theta^{t+1}} = \tau Z_{\theta^t} + (1 - \tau) Z_{\phi^t}$ is introduced. Does the EMA update in the parameter space directly translate to the same update rule in the representation space?

---

> ### Author Response · Authors · 2025-12-03
>
> **Weakness 1:**
>
> **zrcp:** "Insufficient discussion of related work: The paper claims several key contributions, including theoretical analyses of the predictor, stop-gradient mechanism, and EMA, explanations for the non-collapsing behavior of TS-SSL, and the introduction of mutual information-based constraints. However, similar arguments have been made in existing literature [1, 2, 3], and the paper fails to explicitly discuss distinctions from these prior works."
>
> **Answer:**
> We thank the reviewer for raising this point. However, it seems that the references [1, 2, 3] were not included in the review, so we would appreciate clarification regarding which works you are referring to.
>
> That said, we emphasize that our contribution is distinct from prior analyses in several ways. Unlike works that primarily introduce entropy-based regularization (e.g., Rodriguez-Galvez et al., 2024), our regularization is theoretically derived from a mutual-information perspective and simultaneously promotes alignment and uniformity—not uniformity alone.
>
> We provide extensive empirical comparisons between our MI-based regularization and uniformity-only counterparts in Tab. 8 (Sec. D.3), Tab. 5 (Sec. D.2), and Tab. 4 (Sec. D.1). Across a wide range of regularization strengths, the best-performing RoBYOL and RoSiam models generally outperform the best BYOL + $U(Z_\phi)$ variants, particularly on medical imaging benchmarks. This supports both the validity and the theoretical relevance of our approach.
>
> ---
>
> **Weakness 2:**
>
> **zrcp:** "Overstated title: TS-SSL encompasses diverse methods, such as the classic DINO. Yet, this paper only analyzes BYOL and SimSiam, which are insufficient to represent the broader TS-SSL framework, making the title overly broad."
>
> **Answer:**
> We appreciate the reviewer’s comment. While TS-SSL includes a variety of methods, our analysis is intentionally restricted to the BYOL–SimSiam family, which captures the two key TS-SSL regimes:
> 1) EMA-based teacher–student learning (e.g., BYOL), and
> 2) Stop-gradient–only learning without EMA (e.g., SimSiam).
>
> These two architectures isolate the core mechanisms we study—predictor, stop-gradient, and the presence or absence of EMA updates—without introducing the additional components used in methods like DINO (centering, sharpening, multi-crop, ViT inductive biases), which are outside our theoretical scope.
>
> Our goal is not to cover the entire TS-SSL landscape, but to provide principled insight into these foundational TS-SSL mechanisms. To avoid any ambiguity, we are willing to adjust the title accordingly. We suggest:
> *"Understanding BYOL and SimSiam Under The Lens Of Mutual Information Maximization"*
>
> ---
>
> **Weakness 3:**
>
> **zrcp:** "Limited experimental validity: Experiments are conducted on small-scale datasets, which lack representativeness in the current era. This raises concerns about the method’s effectiveness in real-world application scenarios."
>
> **Answer:**
> We recognize that scaling is important in the SSL literature. Still, Camelyon16, BRACS, and ImageNet100 are substantial benchmarks, widely used in representation-learning research. BRACS in particular is a challenging, large-scale WSI dataset with high clinical relevance.
>
> Our primary objective is to validate the theoretical derivation linking TS-SSL to mutual information and to show that this theoretical view explains the empirical gains obtained by our regularization. The performance improvements observed on medical datasets—where real-world data is often limited—are especially meaningful, as they highlight the robustness and applicability of our method in realistic deployment scenarios (e.g., dermatology, histopathology, hematology, or ophthalmology).
>
> Thus, while scaling to even larger datasets is an important future direction, the chosen benchmarks are well aligned with the objective of the paper: demonstrating that the proposed MI-based regularization is theoretically grounded and practically effective.

---

### Official Review · Reviewer_XwQc · 2025-11-03

**Soundness:** 1
**Presentation:** 1
**Contribution:** 2
**Rating:** 2
**Confidence:** 3

**Summary:**

The paper looks to understand Teacher-Student SSL methods from an information theoretic perspective to justify their performance, analogously to the mutual information maximisation perspectives of other SSL methods, such as infoNCE.

**Strengths:**

The aim of understanding these SSL methods from an information theoretic perspective, interpreting the implicit distributional assumptions they make and making principled improvements is a sound approach.

The material improvement in results on a number of benchmark datasets suggest the proposed method is useful.

**Weaknesses:**

The main weaknesses of the paper are readability and the mathematical arguments don't seem correct or well-presented and I believe need to be materially re-worked.

* readability - the notation is hard to follow (SSL typically considers 2 related samples x, x' and their representation z, z', which is easier to follow than tracking subscripts).
   - The paper overloads symbols and mixes random variables with their realizations. For instance, $Z_\theta=f_\theta(t(X))$ and $Z_\phi=f_\phi(t(X))$ have the same augmentation $t$, then later switch to multi‑view $v_i^{(1)}$, $v_i^{(2)}$ (Eq. (4)). The dependence structure (one view vs two independent views) must be fixed up front, e.g. with $t,t'\stackrel{iid}{\sim}\mathcal{T}$. As written, it implies $Z_\theta, Z_\phi$ are deterministic functions of the same $t(X)$, which is not the BYOL/SimSiam setup and creates confusion for all MI statements.
* The mathematical arguments are difficult to follow, e.g.
   - it would help if the loss function being explained were stated upfront (is this RHS of Eq 5?).
   - definition of conditional entropy (Eq 1) is already an expectation over X (e.g. see https://en.wikipedia.org/wiki/Conditional_entropy)
   - "alignment" is typically considered between representations of related samples, Eq 1 doesn't consider that so hard to see how this relates, e.g. to Wang & Isola.
   - Eqs 2/3 seems to be the Barber & Agakov bound, eg see "On Variational Bounds of Mutual Information" by Poole et
al (as cited!). This is simply an instance of cross entropy lower bounding entropy (Eq 2), which is a fundamental of machine learning and far from a "novelty".
   - the last term in Eq 2 should be in expectation (over X).
   - rather than referring to "kernel density estimation" (vague/general), it should be stated in explicit terms if the first term of Eq 3 is equivalent to Eq 4 under specific Gaussian assumptions for p and q (assuming that is the case)? It is well known that the cross entropy of two Gaussians has the general form in Eq 4 (under specific assumptions that are not made clear here).
   - Eq 4 mixes exact expectations and MC estimates (from samples).
   - Eq 4 is a function of z under two different distributions (where z is a representation of a particular view of x), it is unclear how that then becomes a function of different z's/views.
      - From the outset, it would be clearer to refer to x and x' as different views (or similar), as is common to avoid confusion.
   - it is unclear what happens to the entropy term in Eq 3.
   - in the context of the number of operations in a neural network, it is invalid to suggest that a multiplicative factor is set to 1 for "computational efficiency" (rather this is part of the p/q assumptions above)
* the assumptions (including Eq 11) seem strong, unintuitive or not very well justified, particularly Assumption 2 and Eq 11.

Details
* 268 - t already used to define augmentation
* 269 - Z's are defined as deterministic functions of the same t(X) and therefore of each other? It is unclear why this relationship would be time invariant given that other functions are not.
* 321 - this seems highly unintuitive for a relationship presumed to hold throughout training for a finite model.

Minor
* 177 - include the domain of v = t(x) (presumably $\mathcal{X}$)
* 192 - an unusual way of writing conditional cross entropy (double lines usually reserved for divergence)

**Questions:**

See weaknesses

---

> ### Author Response · Authors · 2025-12-03
>
> **Weakness 1:**
>
> **XwQc:** "readability -- the notation is hard to follow (SSL typically considers two related samples $x, x'$ and their representations $z, z'$, which is easier to follow than tracking subscripts).
> The paper overloads symbols and mixes random variables with their realizations. For instance, $Z_{\theta} = f_{\theta}(t(X))$ and $Z_{\phi} = f_{\phi}(t(X))$ have the same augmentation $t$, then later switch to multi-view $v_i^{(1)}, v_i^{(2)}$ (Eq. 4). The dependence structure (one view vs. two independent views) must be fixed up front, e.g., with $t, t' \overset{\text{iid}}{\sim} \mathcal{T}$. As written, it implies $Z_{\theta}, Z_{\phi}$ are deterministic functions of the same $t(X)$, which is not the BYOL/SimSiam setup and creates confusion for all MI statements."
>
> **Answer:**
> We thank the reviewer for the insightful comments regarding the notation and dependence structure of the representations. We acknowledge that the previous formulation could be interpreted as defining $Z_\theta$ and $Z_\phi$ as deterministic functions of the *same* augmented input $t(X)$, which is not representative of the BYOL/SimSiam setting and may cause confusion in the context of mutual information statements.
>
> To address this issue, we have revised the notation to explicitly reflect the use of *independent* augmentations in self-supervised learning. In particular, we now sample $t, t' \overset{\text{iid}}{\sim} \mathcal{T}$ and define the corresponding augmented views as $v = t(x)$ and $v' = t'(x)$, ensuring that the teacher and student process different views of the same input. The associated representations are now defined as $Z_\theta := f_\theta(t(X))$ and $Z_\phi := f_\phi(t'(X))$, where $t(.)$ (and $t'(.)$) is an augmentation randomly drawn for each image, which makes their stochastic dependence explicit and eliminates the unintended implication that $Z_\theta$ and $Z_\phi$ are deterministic functions of one another.
>
> Furthermore, we clarified the domain of the augmented views by introducing the view space $\mathcal{V}$ and added a convention distinguishing random variables (uppercase, e.g., $Z_\theta$) from their realizations (lowercase, e.g., $z_\theta$). This prevents symbol overloading and ensures that our mutual information expressions are well defined between random variables.
>
> These modifications appear in the revised manuscript and are highlighted in blue. We believe they substantially improve the clarity and readability of the exposition, and we thank the reviewer again for pointing out this important issue.
>
> ---
>
> **Weakness 2:**
>
> **XwQc:** "The mathematical arguments are difficult to follow, e.g. It would help if the loss function being explained were stated upfront (is this RHS of Eq 57?)."
>
> **Answer:**
> We thank the reviewer for this feedback. We added in the main manuscript a paragraph to recall BYOL and SimSiam's loss and how it connects to our equations.

---

> ### Author Response · Authors · 2025-12-03
>
> **Weakness 3:**
>
> **XwQc:** "The mathematical arguments are difficult to follow, e.g.
> - Definition of conditional entropy (Eq 1) is already an expectation over $X$ (e.g. https://en.wikipedia.org/wiki/Conditional_entropy)
> - The last term in Eq 2 should be in expectation (over $X$).
> - Rather than referring to 'kernel density estimation' (vague/general), it should be stated in explicit terms if the first term of Eq 3 is equivalent to Eq 4 under specific Gaussian assumptions for $p$ and $q$ (assuming that is the case)? It is well known that the cross entropy of two Gaussians has the general form in Eq 4 (under specific assumptions that are not made clear here).
> - Eq 4 mixes exact expectations and MC estimates (from samples).
> - Eq 4 is a function of $z$ under two different distributions (where $z$ is a representation of a particular view of $x$), it is unclear how that then becomes a function of different $z$'s/views.
>   - From the outset, it would be clearer to refer to $x$ and $x'$ as different views (or similar), as is common to avoid confusion.
> - It is unclear what happens to the entropy term in Eq 3."
>
> **Answer:**
>
> **Response to Reviewer on Conditional Entropy Notation.**
> We thank the reviewer for the comment regarding the definition of conditional entropy.
> To improve clarity, we have modified Eq. (1) to
>
> $I(X; Z_\theta) = - E_{x \sim p_X} ( H(Z_\theta \mid X = x) ) + H(Z_\theta)$
>
> so that the conditioning on $x$ is explicit. This makes it clear that $H(Z_\theta \mid X=x)$ refers to the entropy of $Z_\theta$ conditioned on a particular input $x$, rather than suggesting that $H(Z_\theta \mid X)$ is the entropy for a single sample. We believe this change improves readability for readers who may not be familiar with the standard expectation-over-$X$ convention in information theory.
>
> **Gaussian Assumptions and KDE Parameterization.**
> The reviewer is correct that the previous reference to "kernel density estimation" was too general. We have now rewritten the corresponding paragraph to make all modeling assumptions explicit. In particular, we state that:
>
> - We draw $L$ samples $z_\theta^{(l)} = f_\theta(v_i^{(l)})$ from the teacher distribution $p_\theta(z \mid X=x_i)$ via independently sampled augmentations $v_i^{(l)} \sim \mathcal{T}$ (re-substitution estimator as in Wang et al., 2020).
> - We estimate $q_{\phi,\psi}(z_\theta \mid x)$ using a Gaussian kernel density estimator with $K$ draws centered at $h_\psi(f_\phi(v_i^{(k)}))$ and identity covariance (as in Louiset et al., 2024, Sec. A.2).
>
> Under these explicit Gaussian assumptions, the Cross-Prediction term takes the closed-form expression in Eq. (6) of the revised manuscript. This resolves the ambiguity raised by the reviewer: we no longer invoke KDE in a vague manner—the Gaussian structure used to derive Eq. (6) of the revised manuscript is now fully stated.
>
> **Monte-Carlo Estimation and Expectations.**
> We acknowledge the reviewer’s concern that Eq. (6) (of the revised manuscript) mixes exact expectations and Monte-Carlo estimates. We have revised the notation to reflect that, in practice, all empirical quantities are approximated via Monte-Carlo sampling over augmented views. Expectations are now written explicitly as $E_{x \sim p_X} \, E_{z_\theta \sim p_\theta(. \mid x)}(.)$, and estimated using finite-sample averages. This makes clear what is theoretical and what is computed in practice.
>
> **Views and Notation Consistency.**
> We agree that introducing $x$ and $x'$ without specifying their relationship to augmented views could lead to confusion. To address this, we have revised the exposition throughout the section to consistently denote $v = t(x)$ and $v' = t'(x)$ as two independently sampled augmented views of the same input $x$. The latent variables $Z_\theta = f_\theta(v)$ and $Z_\phi = f_\phi(v')$ are now introduced explicitly as random variables induced by these views. This modification resolves the ambiguity the reviewer noted, as Eq. (4) now operates on clearly defined latent variables corresponding to distinct augmentations of the same sample.
>
> **Entropy Term in Eq. (3).**
> Regarding Eq. 4 of the revised manuscript, adding the entropy $H(Z_\theta)$ to both sides of the equality yields Eq. 5. Thus, the left-hand side of Eq. 5 becomes equivalent to Eq. 1.

---

> ### Author Response · Authors · 2025-12-03
>
> **Weakness 4:**
>
> **XwQc:** "The mathematical arguments are difficult to follow, e.g.
> - 'alignment' is typically considered between representations of related samples. Eq 1 doesn't consider that so hard to see how this relates, e.g., to Wang & Isola.
> - Eqs 2/3 seems to be the Barber & Agakov bound, e.g., see 'On Variational Bounds of Mutual Information' by Poole et al (as cited!). This is simply an instance of cross entropy lower bounding entropy (Eq 2), which is a fundamental of machine learning and far from a 'novelty'."
>
> **Answer:**
> We use the term *alignment* in two related but distinct senses, which we make explicit to avoid ambiguity.
>
> **View-Alignment** (as in Wang & Isola): given two augmentations $v = t(x)$ and $v' = t'(x)$ of the same input $x$, a single encoder $f_\theta$ should produce similar representations, i.e. $f_\theta(v) \approx f_\theta(v')$. This encourages invariance across augmentations and corresponds to reducing the conditional entropy $H(Z_\theta \mid X)$.
>
> **Teacher–Student Alignment** (or *Cross-Prediction*): in prediction-based SSL methods such as BYOL and SimSiam, two different networks process the two views: the teacher $f_\theta$ encodes $v$, while the student–predictor pair $(f_\phi, h_\psi)$ encodes and predicts $v'$. The objective is to match $f_\theta(v)$ with $h_\psi(f_\phi(v'))$. This is not view-alignment under the same encoder, but *cross-network alignment*, i.e., $f_\theta(t(x)) \approx h_\psi(f_\phi(t'(x)))$.
>
> **On Barber-Agakov bound:**
> We fully agree with the reviewer that the decomposition used in Eqs. (2)--(3) of the original manuscript is a direct instance of the variational mutual information lower bound introduced by Barber & Agakov (2003) and summarized in Poole et al. (2019). We have corrected the manuscript to avoid suggesting otherwise. The theoretical contribution of our work is *not* the derivation of a new mutual information bound. Rather, the novelty lies in showing that BYOL and SimSiam implicitly optimize this bound through their two-step teacher–student training procedure, even though these methods were not initially motivated from an information-theoretic perspective.
>
> **Weakness 5:**
>
> **XwQc:** "The assumptions (including Eq 11) seem strong, unintuitive or not very well justified, particularly Assumption 2 and Eq 11."
>
> **Answer:**
> We thank the reviewer for raising concerns regarding Assumption 2 and Eq. 11. We agree that, as originally stated, these assumptions may appear stronger than necessary. We have revised the manuscript to clarify their scope and practical relevance.
>
> **On Assumption 2.**
> Assumption 2 does *not* require the network to exhibit globally linear behaviour or constant derivatives. Rather, it expresses a *local first-order approximation* of the teacher update induced by the Exponential Moving Average (EMA). In practice, BYOL-style methods use $\tau \ge 0.99$, which ensures that the teacher parameters evolve through exceedingly small steps. This allows $f_{\theta^{t+1}}(x)$ to be accurately approximated by the first-order Taylor expansion around $f_{\phi^t}(x)$, yielding a linear interpolation of outputs. We now explicitly replace the previous NTK argument with this local approximation, which is both standard and empirically justified. Importantly, this assumption is only required in Section 4.1 to derive closed-form expressions—it is *not* needed for the validity of the mutual information framework developed in Section 3.
>
> **On Eq. 11.**
> Eq. 11 provides a *sufficient* condition ensuring that the EMA update increases mutual information. It is not intended as a structural assumption on the model. Under the common isotropic Gaussian approximation of the teacher activations, Eq. 11 reduces to the check:
>
> $\frac{\sigma_{t+1}}{\sigma_t} \ge \tau$
>
> which depends only on the marginal scale of $Z_\theta$. We empirically verify this condition across six datasets and observe that it holds throughout almost the entire training trajectory for momentum values used in practice ($\tau > 0.99$). Thus, while Eq. 11 may seem abstract in isolation, it corresponds to a mild condition that is systematically satisfied by standard self-supervised learning pipelines, and therefore does not limit the applicability of our result.
>
> We have updated the paper accordingly to prevent these points from being interpreted as restrictive modelling assumptions. Rather, they constitute practical and empirically valid approximations that enable a tractable analysis of EMA dynamics in teacher–student SSL.
>
> ---
>
> **Weakness 6:**
>
> **XwQc:** "In the context of the number of operations in a neural network, it is invalid to suggest that a multiplicative factor is set to $1$ for 'computational efficiency' (rather this is part of the $p/q$ assumptions above)."
>
> **Answer:**
> Indeed, the reviewer is right. We will replace the text by saying "for simplicity" and retain "for computational efficiency" only when referring to the number of views ($K = L = 1$).

---

### Official Review · Reviewer_Rsjj · 2025-11-03

**Soundness:** 2
**Presentation:** 3
**Contribution:** 3
**Rating:** 4
**Confidence:** 3

**Summary:**

The authors study teacher-student self-supervised learning (TS-SSL) methods and focus on addressing the issue that these methods lack a clear information-theoretic explanation. They show that TS-SSL implicitly maximizes a lower bound on the mutual information between inputs and the teacher representations. They also give convergence results characterizing the evolution of the teacher representation’s entropy and alignment during training. By introducing a mutual-information–based regularizer on the student latent space, the authors give empirical results that show improvements on natural-image and medical-imaging benchmarks.

**Strengths:**

The authors show theoretically that TS-SSL maximizes a lower bound on the mutual information between inputs and the teacher representations, which helps to design a mutual-information–based regularizer that leads to empirical improvements on real datasets. The results are interesting.

**Weaknesses:**

This manuscript contains some technical weaknesses, such as referring to missing Sections, lacking of justification for important assumptions and inconsistent derivations, which I list in detail in the QUESTIONS part.

**Questions:**

1. In Section 4.1 the authors refer to Sections A.5 and A.6 in the appendix. However, I cannot find Sections A.5 and A.6 in the manuscript.

2. In Assumption 2, the function f is assumed to exhibit first-order variation (i.e., the derivative remains constant), which is claimed to hold for both the teacher and the student terms. This assumption is somewhat too strong. Please justify. Furthermore, the parameter variation is intertwined with the variation of the mutual information term (Eq. 12) in Lemma 1, which serves as a core of the analyses and impacts the validity of the conclusion.

3. From Lemma 1 and Eq. 9 in Section 3.1, the authors analyze the variation of the teacher term given the student term and input indicating that this conditional entropy can be used to assess the lower bound of mutual information. However, the conditional entropy in Eq. 16 does not incorporate the student term as a condition. Instead it directly asserts that the entropy conditioned solely on the input should be reduced. This is somewhat inconsistent with the previous derivations.

4. Regarding Eq. 17, the authors’ explanation of this equation can be insufficient. They do not clarify the rationale for introducing the entropy constraint in Eq. 17. In the preceding content, the authors illustrate that the improvement direction involves reducing the conditional entropy and increasing the marginal entropy, but they fail to explain why the conditional entropy constraint is specifically introduced here.

5. Regarding Eq. 18, the authors previously mentioned the need to reduce one variance ratio and increase another. However, in Eq. 18, both regularization terms are assigned coefficients greater than zero, which is strange.

6. Typos/small mistakes:

In Eq. (18), is the first Lagrangian multiplier term \lambda H(Z_{\phi})?

In Table 1, ROBYOL is actually lower than BYOL on IN100 (R50).

In Abstract, SSL is used without definition.

---

> ### Author Response · Authors · 2025-12-03
>
> ### Question 1: Missing Appendix Sections
>
> **Rsjj:** “In Section 4.1 the authors refer to Sections A.5 and A.6...”
>
> **Answer:** We apologize for the broken references. The relevant proofs and details are located in **Appendix C.3 and C.4**. We have fixed these citations in the updated version of the paper.
>
> **Question 2: On the validity of Assumption 2**
>
> **Rsjj:** "In Assumption 2, the function f is assumed to exhibit first-order variation (i.e., the derivative remains constant), which is claimed to hold for both the teacher and the student terms. This assumption is somewhat too strong. Please justify. Furthermore, the parameter variation is intertwined with the variation of the mutual information term (Eq. 12) in Lemma 1, which serves as a core of the analyses and impacts the validity of the conclusion."
>
> **Answer:** We appreciate the reviewer's comment regarding the strength of Assumption 2. The reviewer is correct that the implicit assumption of a constant derivative over a large region is too strong. We believe that a justification based on the empirically-enforced small step size of the Exponential Moving Average (EMA) update is more robust and generally applicable. We will revise the manuscript to replace the NTK-based justification for Assumption 2 with the following first-order Taylor approximation:
>
> **Revised Justification for Assumption 2:**
> The linear interpolation of outputs can be assumed using the fact that the EMA coefficient $\tau$ is chosen as close to $1$ ($\tau \geq 0.99$ is typical in practice). This ensures the teacher network's weights move by only a minimal amount, allowing the non-linear network function $f_{\theta}(x)$ to be locally approximated by its first-order Taylor expansion.
>
> **Small Step Size and Linear Approximation:**
> The teacher weight update is: $$\theta^{t+1} = \tau \theta^t + (1-\tau) \phi^t$$
>
> We define the change in parameters from the student's perspective to the new teacher's weights: $$\Delta\theta' = \theta^{t+1} - \phi^t$$
> Substituting the EMA rule: $$\Delta\theta' = \left( \tau\theta^t + (1-\tau)\phi^t \right) - \phi^t$$
> $$\Delta\theta' = \tau\theta^t - \tau\phi^t = \tau (\theta^t - \phi^t)$$
>
> Since the network output $f_{\theta}(x)$ is a differentiable function of the parameters $\theta$, we use the **first-order Taylor expansion** of the new teacher output $f_{\theta^{t+1}}(x)$ around the student parameters $\phi^t$:
> $$f_{\theta^{t+1}}(x) \approx f_{\phi^t}(x) + \nabla_{\phi^{t}} f_{\phi^{t}}(x)^\top (\theta^{t+1}-\phi^{t}) \quad \text{(Eq. 1)}$$
>
> **Isolating the Gradient Term:**
> Next, we approximate the previous teacher output $f_{\theta^{t}}(x)$ by expanding it around the same student parameters $\phi^t$:
> $$f_{\theta^{t}}(x) \approx f_{\phi^t}(x) + \nabla_{\phi^{t}} f_{\phi^{t}}(x)^\top (\theta^{t}-\phi^{t}) \quad \text{(Eq. 2)}$$
>
> By rearranging the first-order part of Eq. 2, we can isolate the gradient term:
> $$\nabla_{\phi^{t}} f_{\phi^{t}}(x)^\top (\theta^{t}-\phi^{t}) \approx f_{\theta^{t}}(x) - f_{\phi^t}(x)$$
>
> **Derivation of Linear Interpolation:**
> Substitute the expression for $(\theta^{t+1}-\phi^t) = \tau (\theta^t - \phi^t)$ and the isolated gradient term back into the first-order part of Eq. 1:
> $$f_{\theta^{t+1}}(x) \approx f_{\phi^{t}}(x) + \tau \left[ \nabla_{\phi^{t}} f_{\phi^{t}}(x)^\top (\theta^{t}-\phi^{t}) \right]$$
>
> Thus, yielding the desired linear interpolation:
> $$f_{\theta^{t+1}}(x) \approx \tau f_{\theta^{t}}(x) + (1-\tau) f_{\phi^{t}}(x)$$
>
> This explicitly shows that the linear interpolation of the outputs can be approximated using a first-order approximation.
>
> **Approximation Error:**
> The linear interpolation holds because the second-order error terms are negligible. In Eq.1, the Taylor remainder $R_{\theta^{t+1}, \phi^{t}}$ depends on the square of the parameter difference $\|\theta^{t+1}-\phi^{t}\|$:
> $$R_{\theta^{t+1}, \phi^{t}} = O(\|\theta^{t+1}-\phi^{t}\|^2) = O(\tau^2 \|\theta^{t}-\phi^{t}\|^2)$$
>
> In Eq. 2, the Taylor remainder $R_{\theta^{t}, \phi^{t}}$ is proportional to $(1-\tau)^2$, which is close to $0$:
> $$R_{\theta^{t}, \phi^{t}} = O(\|\theta^{t+1}-\theta^t\|^2) = O((1-\tau)^2 \|\phi^t-\theta^t\|^2)$$
>
> For typical EMA values $\tau \geq 0.99$, we have $(1-\tau)^2 \le 10^{-4}$. Because the EMA rule actively keeps $\theta^t$ and $\phi^t$ close throughout training, the magnitude of the difference $\|\phi^t-\theta^t\|$ also remains small. Consequently, the second-order terms can be safely ignored.
>
> **Parameter variation in Eq. 12 (Lemma 1):**
> We thank the reviewer for its remark. However, we wish to clarify that the linear interpolation assumption is only necessary for deriving the formulas in Section 4.1. The core theoretical framework and analysis of mutual information in Section 3 does not require this assumption.

---

> ### Author Response · Authors · 2025-12-03
>
> **Question 3: Objective of Sec. 4**
>
> **Reviewer (Rsjj):**
> "From Lemma 1 and Eq. 9 the authors analyze the variation of the teacher term given the student term and input, but Eq. 16 no longer conditions on the student. This seems inconsistent."
>
> **Answer:**
> The difference comes from the distinct goals of Sec. 3 and Sec. 4. In Sec. 3, we show that BYOL and SimSiam implicitly maximize $I(Z_\theta; (X, Z_\phi))$, a lower bound of the mutual information $I(Z_\theta; X)$, by introducing a student encoder and a prediction head as well as a two-step optimization process, with a stop gradient and EMA (with $\tau$ being either equal to $0$ in SimSiam, and $0.99$ or $0.999$ in BYOL). However, this section does not study how $I(Z_\theta;X)$ evolves over training. Therefore, in Sec. 4, we analyze the *actual incremental change* of the teacher’s mutual information $I(Z_\theta; X)$ under the EMA update.
>
> Under Assumption 2, the teacher update
>
> $$Z_{\theta^{t+1}} = \tau Z_{\theta^t} + (1-\tau) Z_{\phi^t}$$
>
> yields closed-form variance dynamics (Eqs. 16–18 of revised manuscript). In this setting, the student’s influence enters *through the variance terms* $\sigma_{\phi^t}^2$ and $\sigma_{X,\phi^t}^2$, not as an explicit conditioning variable. Thus, Eq. 18 uses $H(Z_\theta \mid X)$, but the student still affects the update via these ratios.
>
> **Motivation for Sec. 4.2:**
> The dynamical analysis shows that teacher entropy and alignment may not grow monotonically, potentially causing MI plateaus or decrease. This motivates the regularizer of Sec. 4.2, which promotes higher $H(Z_\phi)$ and lower $H(Z_\phi \mid X)$, ensuring more consistent MI growth and improved downstream accuracy.
>
> **Question 4: Clarification on Eq. 17**
>
> **Reviewer (Rsjj):**
> Regarding Eq. 17, the authors’ explanation of this equation can be insufficient. They do not clarify the rationale for introducing the entropy constraint in Eq. 17. In the preceding content, the authors illustrate that the improvement direction involves reducing the conditional entropy and increasing the marginal entropy, but they fail to explain why the conditional entropy constraint is specifically introduced here.
>
> **Answer:**
> In Sec. 4.1 (Eq. 18 of the revised manuscript), we show that the incremental dynamics of the teacher *conditional* entropy satisfies:
>
> $$
> H(Z_{\theta^{t+1}} \mid X) - H(Z_{\theta^t} \mid X) = \frac{d}{2} \log \left( \tau^2 + (1-\tau)^2 \frac{\sigma_{X,\varphi^t}^2}{\sigma_{X,\theta^t}^2} + 2 \tau (1-\tau) \rho_X^t \frac{\sigma_{X,\varphi^t}}{\sigma_{X,\theta^t}} \right)
> $$
>
>
> As discussed in the interpretation paragraph of Sec. 4.1, Eq. 16 shows that the teacher becomes better aligned with the student precisely when the ratio $\frac{\sigma_{X,\varphi^{t}}^2}{\sigma_{X,\theta^{t}}^2}$ decreases. Indeed, if the expression inside the logarithm becomes smaller than 1, then $H(Z_{\theta^{t+1}}\mid X) \le H(Z_{\theta^{t}}\mid X)$, meaning that the teacher’s conditional entropy decreases and teacher–student alignment improves.
>
> Thus, minimizing the ratio $\sigma_{X,\varphi^{t}}^2 / \sigma_{X,\theta^{t}}^2$ directly encourages a decrease in the teacher conditional entropy. This is exactly why Eq. 19 (of the revised manuscript, 17 in the original manuscript) introduces the constraint $H(Z_\varphi \mid X) \le H(Z_\theta \mid X)$: it enforces at the student level the sufficient condition that guarantees a reduction of teacher conditional entropy through the EMA update, and therefore improves alignment.
>
> In summary, Eq. 16 shows that *increasing* $\sigma_{\varphi^t}^2 / \sigma_{\theta^t}^2$ increases the teacher *marginal* entropy. And Eq. 18 shows that *decreasing* $\sigma_{X,\varphi^t}^2 / \sigma_{X,\theta^t}^2$ decreases the teacher *conditional* entropy. Both effects contribute to increasing $I(Z_\theta;X)$, which motivates the two entropy constraints in Eq. 19 of revised manuscript.

---

> ### Author Response · Authors · 2025-12-03
>
> **Question 5: Clarification on Eq. 18**
>
> **Rsjj:** "Regarding Eq. 18, the authors previously mentioned the need to reduce one variance ratio and increase another. However, in Eq. 18, both regularization terms are assigned coefficients greater than zero, which is strange."
>
> **Answer:**
> The reviewer correctly notes that Eq. 19 (of the revised manuscript) appears to be inconsistent with the stated goal of (i) increasing $H(Z_\varphi)$ and (ii) decreasing $H(Z_\varphi \mid X)$. This is due to a sign typo: our intention is to *maximize* the student marginal entropy (promoting uniformity) and *minimize* the student conditional entropy (promoting alignment). The correct Lagrangian-regularized objective minimizes $- H(Z_\varphi)$ and minimizes $H(Z_\varphi \mid X)$, in order to increase the marginal variance ratio $\sigma_\varphi^2 / \sigma_\theta^2$ and decreasing the conditional variance ratio $\sigma_{X,\varphi}^2 / \sigma_{X,\theta}^2$. We will update the corresponding equation accordingly in the revised manuscript.
>
> **Question 6: Clarification on BYOL and RoBYOL in ImageNet-100**
>
> **Reviewer (Rsjj):**
> "Typos/small mistakes:
> (1) In Table 1, RoBYOL is actually lower than BYOL on IN100 (R50).
> (2) In the abstract, SSL is used without definition."
>
> **Answer:**
> Thank you for pointing these out.
>
> 1. The BYOL value in Table 1 contained a typo: the correct accuracy is **84.56**, consistent with the previously reported of RoBYOL of +0.16 improvement over BYOL.
> 2. We also corrected the abstract to define "SSL" upon first use.
>
> Both issues have been fixed in the revised manuscript.

---

### Author Response · Authors · 2025-12-03
**General comment**

**General Comment:** We thank the reviewer for the thorough and insightful feedback. We have updated the manuscript accordingly, addressing all points raised. Specifically, we have:

- Corrected typos and minor inconsistencies throughout the text and tables.
- Strengthened the notation and clarified the dependence structure of augmented views and representations.
- Expanded discussions and explanations regarding assumptions, including Assumptions 1 and 2, and clarified their practical justification and scope.
- Highlighted ablation studies and empirical analyses to highlight the contribution and novelty of our MI-based regularization relative to uniformity-only baselines.
- Revised the description and presentation of equations to ensure alignment with the intended theoretical framework and experimental implementation.

We believe these revisions substantially improve clarity, readability, and the rigor of our presentation.

---

### Meta-Review · Area_Chair_DmjS · 2026-01-06

**Summary:**

The paper proposes an information-theoretic framework to explain the success of Teacher-Student SSL methods (like BYOL and SimSiam), arguing that they implicitly maximize a lower bound on mutual information. Based on this, the authors introduce a new regularizer and demonstrate empirical improvements.

**Reviewer Concerns:**

While reviewers find the theoretical direction interesting and the empirical results on specific datasets (including medical imaging) positive, the paper suffers from critical flaws, issues include missing appendix sections (the authors admit in the rebuttal it is a mistake), fundamental mathematical errors in derivations (e.g., sign errors in loss functions), confusing notation, and a lack of large-scale validation (full ImageNet).

**Reviewer Scores:**

The reviewers and the AC appreciate the detailed responses of the authors, since  some equation flaws and some poor notation & definitions, which makes this paper not ready for publication, the overall score is low, and most reviewers do not anticipate the discussions

---

### Decision · Program_Chairs · 2026-01-26

Reject